# Reporter-ChIP-nexus reveals strong contribution of the *Drosophila* initiator sequence to RNA polymerase pausing

Wanqing Shao[1†], Sergio G-M Alcantara[1], Julia Zeitlinger[1,2]*

[1]Stowers Institute for Medical Research, Kansas City, United States; [2]Department of Pathology and Laboratory Medicine, University of Kansas Medical Center, Kansas City, United States

**Abstract** RNA polymerase II (Pol II) pausing is a general regulatory step in transcription, yet the stability of paused Pol II varies widely between genes. Although paused Pol II stability correlates with core promoter elements, the contribution of individual sequences remains unclear, in part because no rapid assay is available for measuring the changes in Pol II pausing as a result of altered promoter sequences. Here, we overcome this hurdle by showing that ChIP-nexus captures the endogenous Pol II pausing on transfected plasmids. Using this reporter-ChIP-nexus assay in *Drosophila* cells, we show that the pausing stability is influenced by downstream promoter sequences, but that the strongest contribution to Pol II pausing comes from the initiator sequence, in which a single nucleotide, a G at the +2 position, is critical for stable Pol II pausing. These results establish reporter-ChIP-nexus as a valuable tool to analyze Pol II pausing.
DOI: https://doi.org/10.7554/eLife.41461.001

*For correspondence:
jbz@stowers.org

Present address: †Department of Genetics, The Edison Family Center for Genome Sciences & Systems Biology, Washington University School of Medicine, St Louis, United States

## Introduction

RNA polymerase II (Pol II) pausing is a key regulatory step in metazoan gene regulation. After initiating transcription at the transcription start site (TSS), Pol II often pauses 30–50 bp downstream of the TSS, before being released by p-TEFb into productive elongation (*Adelman and Lis, 2012*; *Gaertner and Zeitlinger, 2014*; *Yamaguchi et al., 2013*). Genome-wide profiling of Pol II and nascent transcripts suggest that pausing is widespread across the genome (*Core et al., 2008*; *Muse et al., 2007*; *Nechaev et al., 2010*; *Zeitlinger et al., 2007*), and inhibition of p-TEFb blocks the productive elongation of the majority of active genes (*Chao and Price, 2001*; *Jonkers et al., 2014*; *Ni et al., 2008*). Together, these results suggest that pause release is a general and obligatory step in transcription.

Despite evidence that pausing is a common feature of transcription, the stability of paused Pol II varies widely between genes. While the degree of pausing may be influenced by transcriptional activation (*Buckley et al., 2014*; *Danko et al., 2013*; *Min et al., 2011*; *Rahl et al., 2010*), differences in Pol II pausing across the genome are, to a large extent, a property of the promoter. In the early *Drosophila* embryo, Pol II pausing is most frequently found at developmental control genes, and a large portion of these genes remain highly paused throughout embryonic development independently of their expression (*Gaertner et al., 2012*; *Zeitlinger et al., 2007*). The promoter sequences of such highly paused genes are notably enriched for specific core promoter elements (*Hendrix et al., 2008*), and promoter swapping experiments demonstrate that Pol II pausing depends on the promoter sequence (*Lagha et al., 2013*). However, despite the evidence that Pol II pausing depends on the promoter sequence, how core promoter elements regulate Pol II pausing remains elusive.

*Drosophila* is ideally suited to study the relationship between promoter sequences and Pol II pausing since a large fraction of promoters in the *Drosophila* genome have clearly defined core

 

promoter sequences that are associated with strong Pol II pausing (*Chen et al., 2013*; *Gilchrist et al., 2010*; *Hendrix et al., 2008*; *Juven-Gershon and Kadonaga, 2010*; *Kwak et al., 2013*; *Lee et al., 1992*; *Vo Ngoc et al., 2017b*). Moreover, these highly paused promoters typically undergo focused initiation, during which Pol II begins transcription at a defined genomic position within a window of a few nucleotides (*Hoskins et al., 2011*; *Kwak et al., 2013*; *Ni et al., 2010*). These promoters are easier to study than those with dispersed initiation, where Pol II initiation occurs throughout a genomic region of ~100 bp. Finally, these highly paused promoters do not typically show strong +1 nucleosomes near the pause site, making Pol II pausing less likely to be dependent on chromatin context (*Benjamin and Gilmour, 1998*; *Bondarenko et al., 2006*; *Brown et al., 1996*; *Gaertner et al., 2012*; *Gilchrist et al., 2010*; *Izban and Luse, 1991*; *Jimeno-González et al., 2015*; *Kwak et al., 2013*; *Rach et al., 2011*).

These convenient and easy-to-study highly paused promoters in *Drosophila* have features that are similar to those of mammalian promoters. Their core promoter elements are recognized by the basal transcription factor TFIID, a highly conserved multi-subunit complex whose promoter binding specificity appears to be conserved across metazoans. For example, recent cryo-EM structures of TFIID show human TFIID bound to all core promoter elements of the synthetic *Drosophila* super core promoter, including the sequences enriched among highly paused promoters (*Louder et al., 2016*; *Patel et al., 2018*). This suggests that the highly paused promoters in *Drosophila* have more elaborate sequence elements, yet rely on conserved metazoan transcription machinery for their function.

TFIID interacts with sequence elements at three regions along the core promoter, which are spaced in a fixed distance from each other (*Juven-Gershon and Kadonaga, 2010*; *Louder et al., 2016*; *Patel et al., 2018*; *Vo Ngoc et al., 2017b*). The first contact occurs at the TATA box, a well-studied core promoter element that is located ~30 bp upstream of the transcription start site and is bound by TATA-binding protein (TBP), a component of TFIID (*Chasman et al., 1993*; *Kim et al., 1993a*; *Kim et al., 1993b*). The second contact occurs at the initiator (Inr) sequence, which overlaps the nucleotide where Pol II initiates transcription (referred to as the +1) (*Chalkley and Verrijzer, 1999*; *Emami et al., 1997*). The Inr is the most common core promoter element in metazoans and is often the only identifiable core promoter element in mammalian promoters (*Hoskins et al., 2011*; *Ohler et al., 2002*; *Smale and Kadonaga, 2003*; *Vo Ngoc et al., 2017a*). The third DNA region contacted by TFIID lies ~30 bp downstream of the transcription start site. It contains short sequence motifs such as the downstream promoter element (DPE), motif ten element (MTE), and the pause button (PB) (*Burke and Kadonaga, 1996*; *Hendrix et al., 2008*; *Kutach and Kadonaga, 2000*; *Lim et al., 2004*; *Purnell et al., 1994*). Despite extensive studies on TFIID-promoter interactions, how core promoter elements affect Pol II pausing is not clear.

Measurements of paused Pol II across the genome, including with ChIP-seq, normally do not assess the duration of pausing directly, but rather represent steady-state Pol II occupancy across a population of cells, in which paused Pol II turns over at different rates. To measure the stability of paused Pol II more directly, a time-course analysis in response to triptolide treatment can be performed. Triptolide inhibits initiation globally and prevents new Pol II from reaching the pause position (*Jonkers et al., 2014*; *Titov et al., 2011*). After triptolide treatment, previously existing paused Pol II is lost over time, either by transitioning into elongation or due to premature transcript termination. The decay rate of paused Pol II under these conditions is then proportional to the stability of paused Pol II (*Henriques et al., 2013*; *Jonkers et al., 2014*; *Krebs et al., 2017*).

We recently performed such time-course measurements of paused Pol II across the *Drosophila melanogaster* genome using a high-resolution exonuclease-based ChIP-seq protocol (ChIP-nexus). This assay has the advantage of distinguishing Pol II signal at the site of initiation and pausing, and thus the Pol II half-life calculations are based on paused Pol II, rather than total Pol II at the promoter (*Shao and Zeitlinger, 2017*). These experiments confirmed the large differences in pausing stability across promoters and showed significant correlations with core promoter elements. However, whether these promoter sequences directly regulate Pol II pausing or whether they are statistically over-represented among promoters with high or low Pol II pausing stability is not known.

A better understanding of the relationship between the core promoter sequence and pausing behavior is difficult to obtain with current techniques. On one hand, traditional biochemical or reporter gene assays usually use gene expression and not Pol II occupancy as the readout since Pol II pausing was not yet known to be a general regulatory step in transcription when these assays were

developed. As a result, in vitro assays for studying Pol II pausing are very limited, and it is not even clear whether Pol II pausing requires a natural chromatin context (*Benjamin and Gilmour, 1998*; *Bondarenko et al., 2006*; *Brown et al., 1996*; *Izban and Luse, 1991*). On the other hand, Pol II pausing is readily detected by genomics techniques in vivo, but manipulating endogenous genomic promoter sequences to analyze their links to Pol II pausing is time consuming. Therefore, having an assay that measures Pol II pausing outside the normal genomic context would allow quick promoter mutagenesis and greatly facilitate studying mechanistic features in the regulation of Pol II pausing.

Although traditional reporter assays were not designed to detect Pol II pausing, it is nevertheless possible that Pol II pausing occurs in some of these assays (*Benjamin and Gilmour, 1998*). We reasoned that if paused Pol II occurred on a plasmid, it would be possible to combine the efficiency of plasmid-based sequence mutagenesis with Pol II ChIP-nexus to rapidly test how alterations in promoter sequences affect Pol II pausing. Unexpectedly, we found that paused Pol II can easily be detected on a plasmid by ChIP-nexus and its footprint closely recapitulates the Pol II footprint at the endogenous locus. Using this novel assay, termed reporter-ChIP-nexus, we analyzed the contribution of various core promoter elements and discovered that the Inr sequence, when containing a G at position +2, plays an unexpectedly large and dominant role in stabilizing paused Pol II. Taken together, we show that reporter-ChIP-nexus combines the ease of plasmid-based sequence mutagenesis with high-resolution ChIP profiling, thereby serving as a valuable tool for dissecting the role of specific sequences in Pol II pausing.

## Results

### Promoter-specific Pol II pausing properties are recapitulated on the reporter

To test whether Pol II pausing is recapitulated on a plasmid, we used a GFP reporter, which has moderate expression when transfected into *D. melanogaster* Kc167 cells and allows rapid insertion of any type of promoter sequence (*Figure 1A* and *Figure 1—figure supplement 1*). After transient transfection of the reporter construct, the entire cell extracts were used for ChIP-nexus since there is no straightforward method for isolating plasmids from fixed cells before chromatin immunoprecipitation. Due to the high number of plasmids per cell, we found that only moderate sequence coverage (~5 to 10 million unique reads per sample) was sufficient to obtain high ChIP signal from the plasmid. Sequencing the endogenous genome in addition to the plasmid also provided a convenient internal control for the ChIP quality and for normalizing samples with each other.

To explore whether Pol II pausing can be detected on this plasmid, we first cloned in the super core promoter (SCP) (*Figure 1B* and Table S1). This synthetic promoter contains correctly positioned TATA, Inr, MTE, DPE and PB core promoter elements such that they are recognized by TFIID and efficiently direct Pol II transcription (*Juven-Gershon et al., 2006*; *Louder et al., 2016*). Since it is a synthetic sequence, it can be unambiguously distinguished from the endogenous promoters in the genome. After transfection of the plasmid, we used ChIP-nexus to map Pol II on this promoter. Interestingly, we observed a strong accumulation of paused Pol II signal at the expected pausing position, suggesting that Pol II can pause on the plasmid (*Figure 1B*).

To validate that transcription on this promoter indeed begins at the initiator sequence and thus the accumulation of Pol II occurs downstream of transcription start site, we developed a gene-specific 5' RNA sequencing protocol (*Figure 2—figure supplement 2*). This assay mapped the vast majority of the 5' ends of the transcribed RNA to the initiator sequence (*Figure 1B*), confirming that the SCP functions as expected.

To test whether the Pol II pausing profile on the plasmids recapitulates the pattern of endogenous promoters, we then cloned promoter sequences from *Drosophila pseudoobscura* into the reporter. *D. pseudoobscura* promoters are sufficiently different from those in the *melanogaster* genome (due to ~25 millions of years of divergence), thereby facilitating the unambiguous mapping of Pol II signal on the plasmid.

Since the promoter sequences might have diverged in function between *D. melanogaster* and *D. pseudoobscura,* we first performed Pol II ChIP-nexus analysis on a *D. pseudoobscura* cell line (ML83-63) and analyzed the stability of paused Pol II after 1 hr triptolide treatment. After 1 hr triptolide treatment, promoters with a Pol II half-life of ~5–10 min show strongly reduced Pol II pausing at the

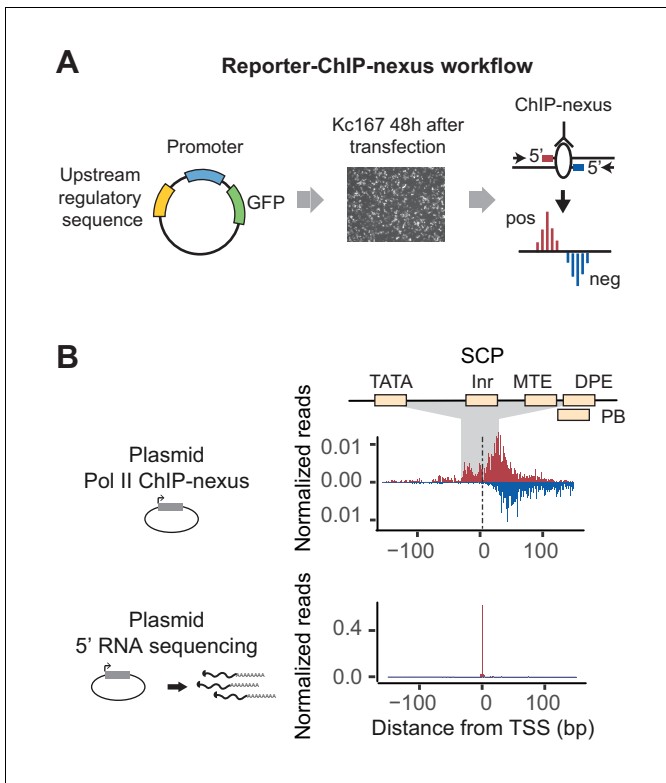

**Figure 1.** Reporter-ChIP-nexus captures paused Pol II. (**A**) Reporter-ChIP-nexus is performed by cloning *Drosophila pseudoobscura* promoters or synthetic promoters into a simple GFP reporter and transfecting into *D. melanogaster* Kc167 cells. The whole cell lysate from cross-linked cells is used to perform Pol II ChIP-nexus. Exonuclease stop bases are then mapped and shown on the positive strand in red above the line, while reads from the negative end are shown in blue below the line. (**B**) Results of reporter-ChIP-nexus reveal strong Pol II pausing at the synthetic super core promoter (SCP), which contains the core promoter elements TATA, Inr, MTE, DPE and PB (top). The position of transcriptional initiation is mapped by sequencing the 5' end of the produced RNA (bottom). The results for the SCP promoter show that the vast majority of RNAs start at the expected site of initiation.

DOI: https://doi.org/10.7554/eLife.41461.002

The following figure supplements are available for figure 1:

**Figure supplement 1.** A simple GFP reporter that allows the fast insertion of any promoter sequence.
DOI: https://doi.org/10.7554/eLife.41461.003

**Figure supplement 2.** Workflow for gene-specific 5' RNA sequencing, a method similar to RNA amplification of cDNA ends (RACE).
DOI: https://doi.org/10.7554/eLife.41461.004

pause position and often show increased levels of Pol II at the site of initiation as previously observed (*Erickson et al., 2018*; *Krebs et al., 2017*; *Shao and Zeitlinger, 2017*). In contrast, stably paused promoters maintain high levels of Pol II at the pausing position with no noticeable increase of Pol II at the site of initiation. This allowed us to identify promoters with strong differences in Pol II pausing for further analysis (*Figure 2—figure supplement 1*).

In total, we selected eight *pseudoobscura* promoters of ~250 bp in length that had a variety of known core promoter elements bound by TFIID (Table S1). We transfected each plasmid into *D. melanogaster* Kc167 cells and mapped the initiation start sites using gene-specific 5' RNA sequencing (*Figure 1—figure supplement 2*). On all plasmids, the start sites mapped within a narrow window of a few base pairs to the predicted Inr sequence (*Figure 2A* and *Figure 2—figure supplement 2*).

We then performed Pol II ChIP-nexus on the transfected cells and compared the Pol II profile on each plasmid to the endogenous profile in *D. pseudoobscura* cells. Again, we detected strong Pol II signal on all plasmids, precisely at the canonical pausing position. Moreover, the Pol II profile was in

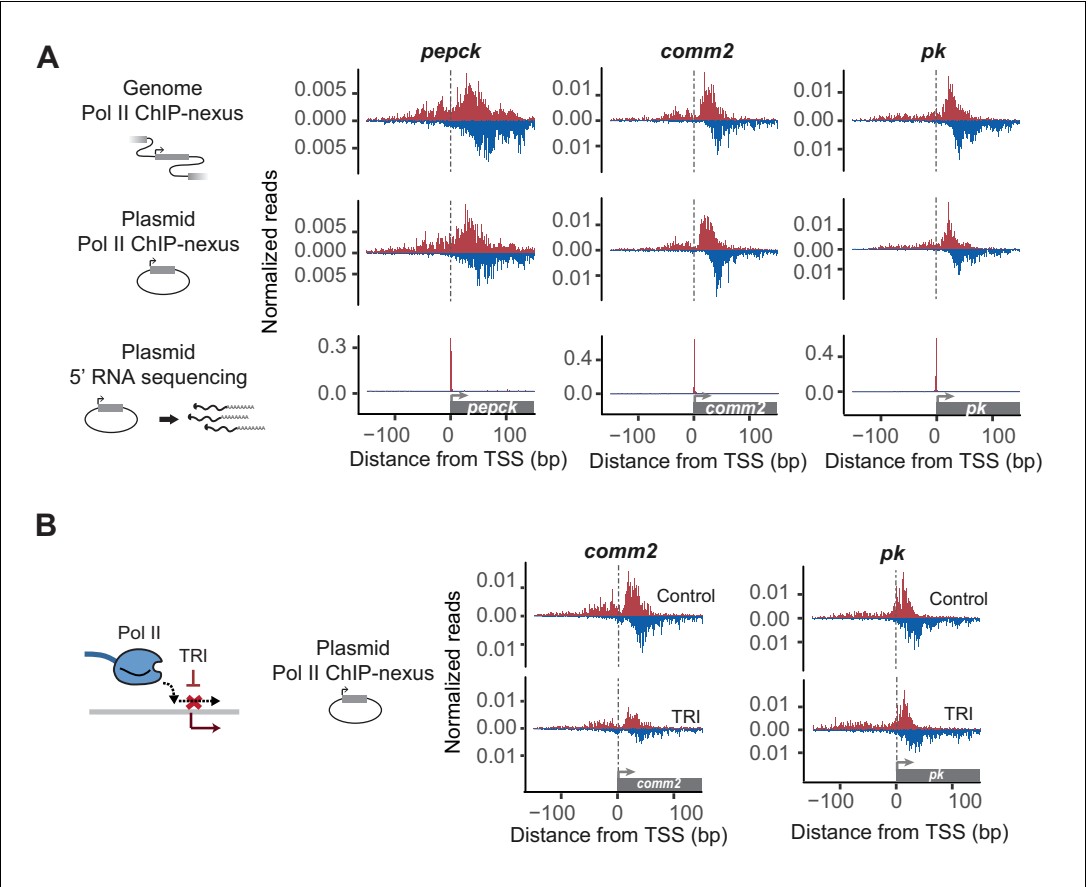

**Figure 2.** Reporter-ChIP-nexus recapitulates the endogenous Pol II pausing profile. (**A**) To obtain the endogenous Pol II pattern of *D. pseudoobscura* promoters in the genome, ChIP-nexus was performed in a *D. pseudoobscura* cell line. Results are shown for the *pepck*, *comm2* and *pk* promoters. The same promoters were then examined using reporter-ChIP-nexus in *D. melanogaster* Kc167 cells, which yielded patterns very similar to the endogenous Pol II profiles. The transcription initiation sites were confirmed by 5' RNA sequencing. (**B**) To determine the stability of paused Pol II on the plasmid, transfected cells were treated with either DMSO (Control) or triptolide (TRI) for 1 hr. The results show a relative reduction in Pol II similar to that of the endogenous loci after treating the *D. pseudoobscura* cell line with TRI (see *Figure 2—figure supplement 4*). Therefore, reporter-ChIP-nexus reveals gene-specific Pol II pausing stability on a plasmid.

DOI: https://doi.org/10.7554/eLife.41461.005

The following figure supplements are available for figure 2:

**Figure supplement 1.** Paused Pol II stability measurements at eight *D. pseudoobscura* promoters.
DOI: https://doi.org/10.7554/eLife.41461.006
**Figure supplement 2.** Reporter-ChIP-nexus recapitulates endogenous Pol II pausing.
DOI: https://doi.org/10.7554/eLife.41461.007
**Figure supplement 3.** Larger promoter region insertion is required for recapitulating Pol II pausing at *RpL13A* on the plasmid.
DOI: https://doi.org/10.7554/eLife.41461.008
**Figure supplement 4.** Reporter-ChIP-nexus recapitulates gene-specific Pol II pausing stability.
DOI: https://doi.org/10.7554/eLife.41461.009

most cases indistinguishable from the endogenous Pol II profile (*Figure 2A* and *Figure 2—figure supplement 2*), with Pearson correlations similar to or slightly below those of replicate experiments (Table S2 and S3). This indicates that lifting a promoter with TFIID-dependent core promoter elements out of its genomic chromatin context does not abolish Pol II pausing and confirms that the pausing profile at those promoters is highly dependent on the promoter sequence.

To test the versatility of our assay, we also tested a *D. pseudoobscura* promoter that is more likely dependent on the chromatin context. The promoter of the ribosomal gene *RpL13A* belongs to the group of promoters that uses TCT as initiator element (*Parry et al., 2010*). These promoters undergo focused initiation, but unlike other focused promoters, have a strong +1 nucleosome with high levels of H3K4me3 (*Figure 2—figure supplement 3*). We found that Pol II pausing can be recapitulated on the plasmid for *RpL13A* when a larger 2 kb region surrounding the core promoter was cloned into the plasmid, but not with a smaller 300 bp region (*Figure 2—figure supplement 3*). The plasmid with the larger region showed high levels of H3K4me3, similar to the endogenous *Rpl13A* promoter, while the construct with the smaller insertion did not (*Figure 2—figure supplement 3*). These results indicate that further technical optimization may be required for examining Pol II pausing on plasmids at promoters where genomic context is essential. This would be consistent with previous studies indicating that plasmids may or may not be correctly chromatinized when transfected into cells (*Jeong and Stein, 1994*; *Reeves et al., 1985*).

Focusing on the set of eight selected promoters, we next tested whether the stability of Pol II after triptolide treatment is recapitulated on the plasmid. To directly compare the Pol II ChIP-nexus profile on the plasmid between triptolide-treated and untreated cells, we used the same pool of plasmid-transfected cells for both samples and normalized them to each other using the genome reads. The results show that paused Pol II is indeed lost after treatment with triptolide and that the degree of loss is proportional to the stability of Pol II on the endogenous promoter. For example, some promoters (e.g. *comm2*) showed a strong reduction of paused Pol II after 1 hr triptolide treatment, while others (e.g *pk*) displayed minimal loss (*Figure 2B* and *Figure 2—figure supplement 4*). Taken together, these results suggest that Pol II not only pauses on the plasmid, but that the promoter-specific stability of paused Pol II can, to some extent, be measured on a plasmid.

## The Pol II pausing stability can be altered by changing the downstream promoter sequence

Strong Pol II pausing correlates with the presence of downstream elements such as the PB, MTE and DPE (*Chen et al., 2013*; *Gaertner et al., 2012*; *Gilchrist et al., 2010*; *Hendrix et al., 2008*; *Kwak et al., 2013*; *Shao and Zeitlinger, 2017*). These sequences may be directly implicated in Pol II pausing since they are located near the site of paused Pol II, and changing the position of the DPE relative to the Inr alters Pol II pausing (*Kwak et al., 2013*). However, further functional evidence for their role in Pol II pausing is lacking, and a detailed mechanistic dissection of these downstream sequences using reporter-ChIP-nexus could be challenging. For example, sequences near the location of paused Pol II would have to be altered, which could indirectly affect the ChIP-nexus profile of Pol II, for example through altered protein-DNA crosslinking efficiency.

We therefore decided to broadly test the role of downstream sequences in Pol II pausing in our initial study. We took two promoters with a short Pol II pausing half-life (*Act5C* and *pepck*) and replaced the entire downstream promoter sequence with that of a stably paused promoter (*pk* or *dve*) (*Figure 3A* and *Figure 3—figure supplement 1*). We then performed Pol II ChIP-nexus on these hybrid promoters with or without triptolide treatment (we treated for 5 min since the wildtype promoters have a short Pol II half-live). While it was difficult to judge whether the wildtype and hybrid promoters had a different Pol II profile under control conditions, we found that a higher fraction of paused Pol II was reproducibly maintained on the hybrid promoters after triptolide treatment compared to the wild-type promoter (*Figure 3B*). Since the loss of Pol II at the pausing position after triptolide treatment reflects the pausing stability, these data suggest that the replacement of the downstream promoter sequence made Pol II pausing more stable.

To quantify the difference in Pol II pausing stability, we first calculated the relative loss of paused Pol II between control and triptolide-treated samples for each construct. Since the same batch of transfected cells are used for this comparison, and the total read counts are normalized to the genomic DNA, the relative loss of paused Pol II is constant and independent of transfection efficiency. Indeed, we found that these measurements were highly reproducible across biological replicates, although the reproducibility was somewhat lower with shorter treatment times of triptolide, presumably because small experimental fluctuations have a larger effect. To compare two constructs, we then determined the difference of triptolide-induced Pol II loss between the mutant construct and the wild-type counterpart. These calculations show that replacing the downstream region in our promoters increased the half-life of paused Pol II around 2-fold (*Figure 3C*).

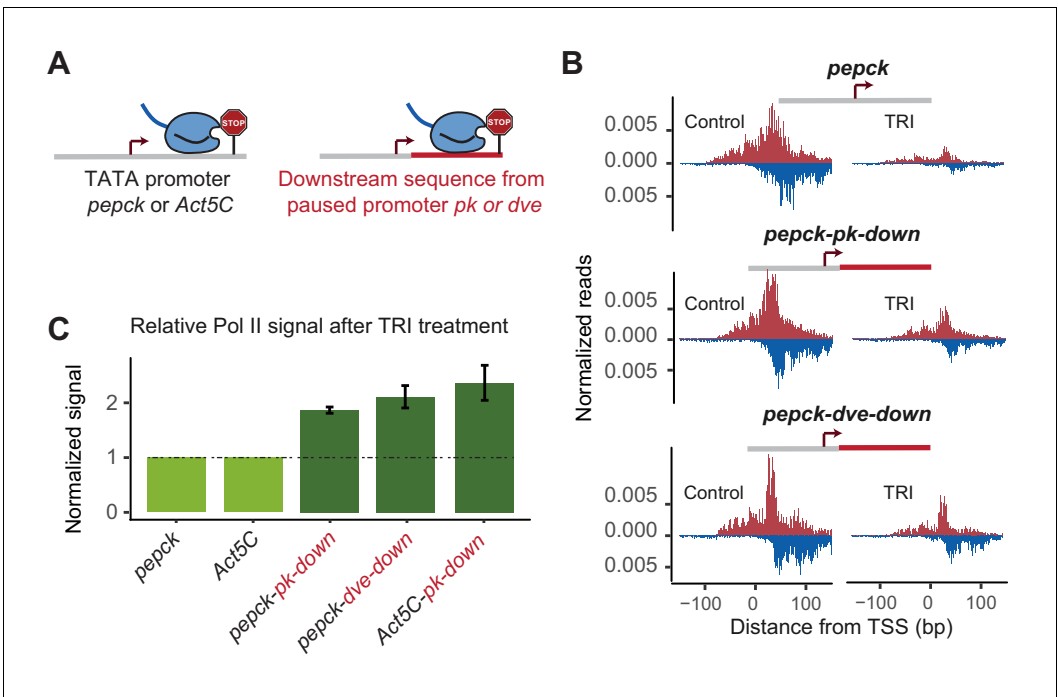

**Figure 3.** Changes in downstream promoter sequences alter paused Pol II stability. (**A**) The *pepck* and *Act5C* downstream sequences were replaced with that from the stably paused promoter *pk* or *dve* (fusion site: 8 bp after the TSS). (**B**) Pol II ChIP-nexus data on the plasmids after transfection into Kc167 cells with or without treatment with triptolide (TRI) for 5 min. The wild-type *pepck* promoter shows a strong reduction of paused Pol II. At the *pepck-pk-down* and *pepck-dve-down* fusion promoters, the same TRI treatment did not reduce paused Pol II to the same extent as at the wild-type promoter, suggesting an increase in the paused Pol II stability as a result of changing the downstream promoter sequence. (**C**) To quantify the difference in paused Pol II stability between different constructs, the ratio of paused Pol II before and after TRI treatment is calculated using two biological replicates. The ratio for the wild-type promoter is then normalized to 1 (light green), and the relative change in this ratio for the fusion promoter is shown on the right (dark green). Error bars refer to the standard error of the mean (TRI treatment: 5 min for all promoters).

DOI: https://doi.org/10.7554/eLife.41461.010

The following figure supplement is available for figure 3:

**Figure supplement 1.** Downstream promoter sequences influences Pol II pausing at *Act5C*.
DOI: https://doi.org/10.7554/eLife.41461.011

---

Taken together, these results confirm that downstream promoter sequences indeed influence the stability of Pol II pausing and we will now refer to MTE, DPE and PB collectively as pausing elements. We next focused on the role of the other TFIID-bound regions in Pol II pausing.

## An upstream region with a TATA box may reduce paused Pol II stability

The TATA box is often enriched among promoters with the lowest amount of Pol II pausing (*Chen et al., 2013*; *Day et al., 2016*; *Gaertner et al., 2012*; *Shao and Zeitlinger, 2017*), and there is evidence in mammalian cells that it promotes the release of paused Pol II (*Amir-Zilberstein et al., 2007*; *Montanuy et al., 2008*). However, some promoters, including the *hsp70* promoter, contain a TATA box and yet show a strong pausing profile (*Buckley et al., 2014*; *Gilchrist et al., 2010*; *Kwak et al., 2013*), questioning the simple model of TATA promoting pause release.

To clarify the correlation between the presence of TATA or other core promoter elements and the stability of Pol II pausing, we first performed a genome-wide data analysis (*Figure 4A* and *Figure 4—figure supplement 1*). We scored each core promoter element based on the presence of consensus sequence at the expected promoter position, allowing for one mismatch: TATA box (STATAWAWR), Inr (TCAKTY), and pausing elements (CSARCSSA, KCGGTTSK or KCGRWCG). We then

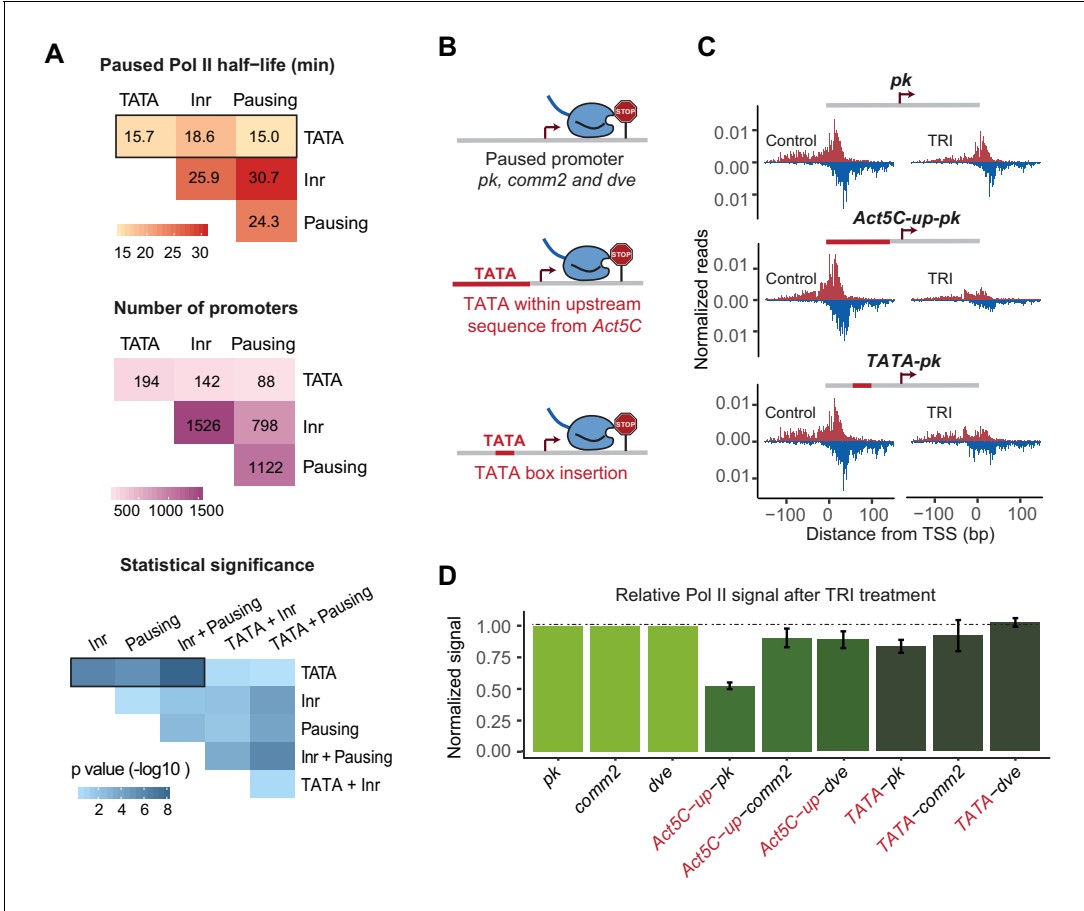

**Figure 4.** A TATA insertion may reduce Pol II pausing. (**A**) Analysis of paused Pol II half-lives as a function of various core promoter element combinations. Median paused Pol II half-life of promoters with different combinations of core promoter elements (top) and the number of promoters classified (middle) are shown. A Wilcoxon rank sum test was then used to test whether the various promoter combinations have significantly different Pol II half-lives from each other (bottom). Core promoter elements other than those used for classification were allowed to occur in each group (e.g. Inr and pausing elements can occur at the 194 promoters with TATA box). Similar results were obtained when no promoter element other than those used for classification were allowed (mutually exclusive model *Figure 4—figure supplement 1*). (**B**) Experimental strategy to functionally test the role of TATA in Pol II pausing: the upstream promoter sequences of the paused promoter *pk, comm2* and *dve* were replaced with that of the TATA promoter *Act5C* (fusion site: 7 bp before the TSS), or a canonical TATA box sequence (TATAAAA) was inserted at 31 bp upstream of the TSS. (**C**) The Pol II ChIP-nexus profiles for wild-type *pk*, the upstream region replacement (*Act5C-up-pk*) and the TATA insertion (*TATA-pk*) under control conditions (left) and after 1 hr treatment with triptolide (TRI, right) show that Pol II pausing is reduced after the *pk* promoter acquired a TATA box. (**D**) Quantification of the relative changes in paused Pol II stability for all fusion promoters relative to the wild-type promoters (TRI treatment: 1 hr for *pk* and *dve* based promoters, 40 min for *comm2* based promoters). Note that only the *pk* promoter shows a clear reduction in Pol II pausing after insertion of a TATA box.

DOI: https://doi.org/10.7554/eLife.41461.012

The following figure supplement is available for figure 4:

**Figure supplement 1.** Correlation between paused Pol II half-life and core promoter elements using a mutually exclusive model.

DOI: https://doi.org/10.7554/eLife.41461.013

analyzed how their presence, alone or in pairwise combination, is correlated with the half-lives of paused Pol II that we measured previously (*Shao and Zeitlinger, 2017*).

As expected, promoters with a pausing element or an Inr tended to have a long Pol II half-life (median ~25 min) and promoters with both elements displayed an even longer half-life (median ~30 min), consistent with them functioning together (*Figure 4A* and *Figure 4—figure supplement 1*). Also consistent with previous findings (*Chen et al., 2013*; *Day et al., 2016*; *Shao and Zeitlinger, 2017*), any combination that contained a predicted TATA box showed a shorter median half-life of

paused Pol II (medians 15–19 min). For example, the median half-life of promoters with a pausing element decreased from ~25 min to ~15 min in the presence of TATA (Wilcoxon $p<10^{-4}$).

To test whether the TATA box robustly destabilized paused Pol II, we then replaced the entire upstream sequence of the TATA-less promoters *pk, comm2* and *dve* with that of the TATA-containing promoter *Act5C* (*Figure 4B*). We reasoned that replacing a larger region will make it more likely that the added TATA box is functional within the larger promoter context. When we then analyzed the stability of Pol II pausing after triptolide treatment (1 hr for *pk* and *dve*, 40 min for *comm2*), we observed reduced Pol II in all constructs as compared to the wild-type promoter (*Figure 4C and D* and *Figure 5—figure supplement 1*). However, the effect was relatively small for the *comm2* and *dve* promoter and only the *pk* promoter showed a strong, almost two-fold, reduction in paused Pol II.

To determine whether this effect was due to the TATA box, we then performed a much smaller alteration and replaced the 7 bp sequence located 31 bp upstream of the transcription start site with a canonical TATA-box sequence (TATAAAA) (*Figure 4B*). We found that in the *pk* promoter (but not the *comm2* and *dve* promoter), this small change indeed reduced the stability of Pol II pausing, albeit to a lesser extent than when the entire upstream region was replaced (*Figure 4C and D* and *Figure 5—figure supplement 1*). This suggests that the TATA box may indeed play a role in destabilizing paused Pol II, but that the overall promoter context plays an important role, too.

## The Inr strongly contributes to the stability of Pol II pausing

To better understand how the TATA box may promote pause release in some promoter contexts, we turned our attention to the Inr. The Inr functions synergistically with the TATA box in transcription if the two elements are optimally spaced from each other (*Emami et al., 1997*; *Malecová et al., 2007*; *O'Shea-Greenfield and Smale, 1992*; *Xu et al., 2011*), but on the other hand, the Inr is overall more highly enriched among highly paused genes (*Gilchrist et al., 2010*; *Hendrix et al., 2008*). This raises the question whether the Inr functions together with TATA in pause release or whether it plays a role in Pol II pausing together with the pausing elements.

To test whether an upstream TATA-containing region is dependent on the Inr, we took the *dve* promoter, which only weakly responded to the TATA-region replacement and swapped both the upstream region and the Inr with the sequences from the TATA-containing *Act5C* promoter (*Figure 5A*). Indeed, this replacement resulted in a strong, more than three-fold reduction in the stability of Pol II pausing (*Figure 5B and C*), suggesting that the type of Inr made a critical difference in this promoter context. However, even when we only replaced the Inr and not the upstream region in the *dve* promoter, we also observed a strong reduction in Pol II pausing stability (*Figure 5B and C*). These results suggest that the Inr sequence itself is an important determinant of Pol II pausing, even independently of the upstream TATA box.

## Highly paused promoters and TATA promoters contain different Inr variants

The observation that the Inr sequence itself is important for pausing prompted us to perform a genome-wide search for Inr variants that may preferentially promote or destabilize Pol II pausing. For this purpose, we compared a strict set of naturally occurring TATA-containing promoters that have a relatively short paused Pol II half-life (<30 min) with those of stably paused promoters without a TATA box (half-life >= 60 min). The results revealed a significant difference in the Inr sequence between the two promoter types (*Figure 6A and B*). The Inr sequence surrounding the first transcribed base of stably paused promoters is best described by the motif TYAGTY (*Figure 6B* left).

In contrast, Inr sequences of TATA-containing promoters are more degenerate with frequent mismatches to the Inr consensus sequences (*Figure 6B* right). These mismatches may even occur at the A, which is the first transcribed base (the +1). However, the most striking difference is that the consensus Inr of stably paused promoters contains a G at the next position (the +2). Stably paused promoters contain a G in 90% of cases, while TATA-containing promoters have a G at this position in only 26% ($p<10^{-46}$). This raises the possibility that the G at the +2 position of the Inr, which we refer to as Inr-G, plays an important role in stabilizing Pol II pausing.

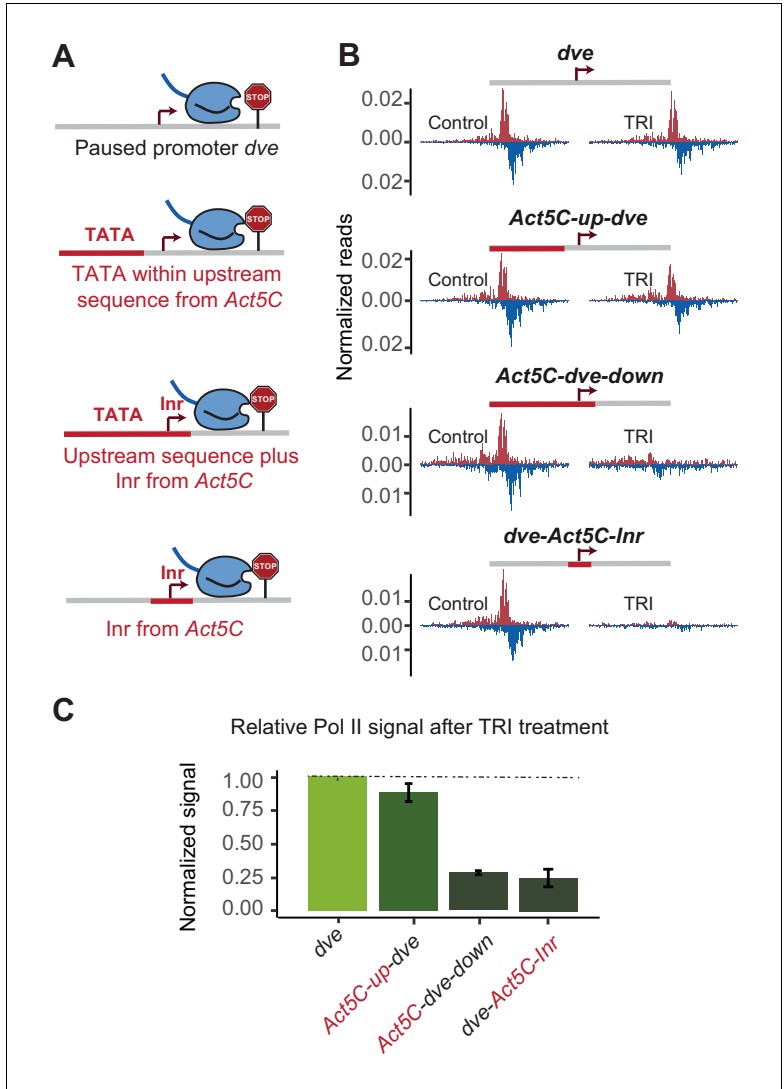

**Figure 5.** The initiator sequence plays an important role in Pol II pausing. (**A**) Experimental series on the *dve* promoter testing the role of the Inr in conjunction with a TATA-containing region. Either only the upstream region was replaced with that of the *Act5C* promoter (fusion site 7 bp before the TSS), the upstream region and the Inr were replaced (fusion site 8 bp after the TSS), or only the Inr was replaced (16 bp region around the TSS). (**B**) Pol II ChIP-nexus profiles under control conditions and after treatment with triptolide (TRI) for 1 hr show a strong reduction of paused Pol II after replacing the Inr sequence. (**C**) Quantification of the relative changes in paused Pol II stability for all fusion promoters relative to the wild-type promoter (TRI treatment: 1 hr for all the promoters).
DOI: https://doi.org/10.7554/eLife.41461.014

The following figure supplement is available for figure 5:

**Figure supplement 1.** Effect of TATA insertion at *comm2* and *dve*.
DOI: https://doi.org/10.7554/eLife.41461.015

## The Inr-G variant plays a dominant role in stabilizing Pol II pausing

The discovery of the Inr-G variant prompted us to revisit our earlier analysis on how different combinations of promoter elements correlate with the half-life of paused Pol II. Strikingly, when we distinguished between Inr-G and Inr-nonG variants, the presence of the Inr-G variant correlated by far the strongest with Pol II pausing (*Figure 7A and B* and *Figure 7—figure supplement 1*). Promoters with an Inr-G variant had a median Pol II half-life of ~44 min, compared to ~14 min for promoters with the Inr-nonG variant (Wilcoxon $p<10^{-48}$) or ~18 min for promoters with a pausing element (Wilcoxon $p<10^{-24}$). When the Inr-G variant was found in combination with a pausing element, the

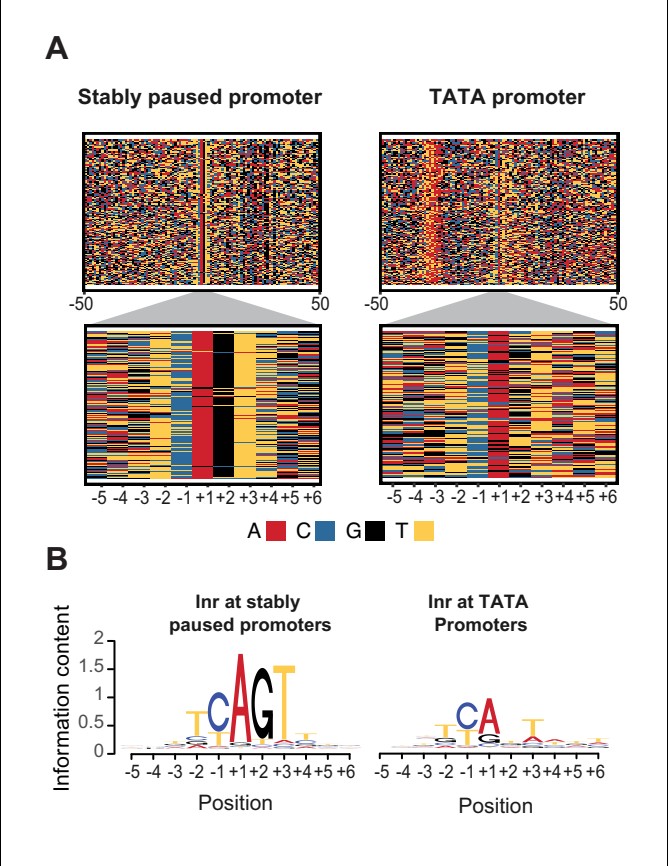

**Figure 6.** Initiator sequences differ between TATA and stably paused promoters. (**A**) The sequence of stably paused promoters (132 randomly selected from the 490 promoters that did not have a TATA box and had a paused Pol II half-life longer than 60 min) are shown on the left as colored letters for a 100 bp window centered on the transcription start site (top). TATA promoters (132 promoters with paused Pol II half-lives shorter than 30 min) are shown on the right as comparison. A higher magnification of the initiator (Inr) sequences below shows clear differences between the two promoter types. (**B**) The consensus motif for the Inr sequences of stably paused promoters match more closely the canonical Inr consensus sequence (TCAKTY) and preferentially contains a G at the +2 position in most cases, whereas the Inr sequences of TATA promoters are more degenerate without enrichment of a G at the +2 position.

DOI: https://doi.org/10.7554/eLife.41461.016

median Pol II half-life was almost 60 min (compared to ~14 min for the Inr-nonG variant, Wilcoxon p<$10^{-62}$). This suggests that while the pausing elements contribute to Pol II pausing, they strongly depend on the Inr-G variant, which overall has the strongest effect on Pol II pausing.

Interestingly, the presence of TATA was still correlated with shorter Pol II half-lives (median ~13 min), but the variant of Inr that was present made a strong difference (*Figure 7A* and *Figure 7—figure supplement 1*). While promoters with a combination of TATA and Inr-nonG had slightly shorter half-lives (median ~11 min), TATA in combination with the Inr-G variant showed very stable Pol II pausing (median ~59 min, Wilcoxon p<$10^{-7}$). This suggests that the Inr acts dominantly over TATA in its effect on Pol II pausing, which explains our earlier observations that introducing a TATA upstream region alone only had a small effect on stably paused promoters.

These results strongly suggest that the G at the +2 position of the Inr is critical for stable Pol II pausing. To validate this experimentally, we specifically mutated the G into A or T at three stably paused promoters (*dve, pk* and the synthetic promoter SCP) and performed Pol II ChIP-nexus under control and triptolide treated conditions (1 hr treatment for *dve* and *pk*, 30-min treatment for SCP) (*Figure 7B–D* and *Figure 7—figure supplement 2*). Importantly, our mutations did not reduce the

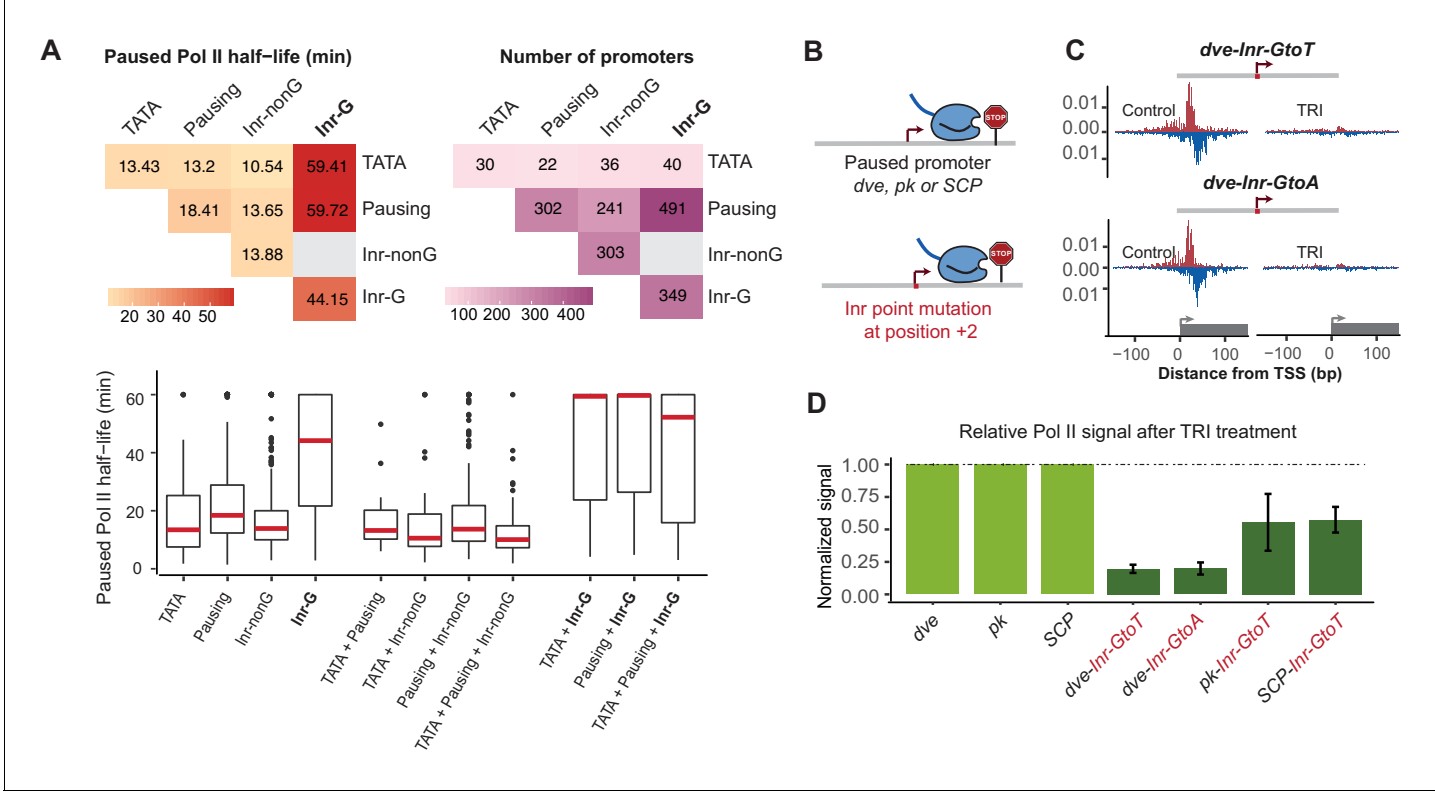

**Figure 7.** The G at Inr + 2 position is critical for stable Pol II pausing. (A) Analysis of paused Pol II half-lives as a function of core promoter element combinations after separating the Inr sequences into those that contain a G at position +2 (Inr-G) versus those that do not contain a G at this position (Inr-nonG). Median paused Pol II half-life (left), promoter numbers (right) and boxplot of their distribution (bottom, median in red) are shown for promoters with different combinations of core promoter elements. The promoters only contain the indicated promoter element, excluding any other promoter element (e.g. the 491 promoters with pausing elements and Inr-G variant do not contain TATA box). Similar results were obtained using a non-mutually exclusive model (*Figure 7—figure supplement 1*). Note that strong Pol II pausing was observed in all combinations that contain the Inr-G variant. Results from testing between combinations using the Wilcoxon rank test are shown in Table S4. (B) Experimental strategy to test the effect of mutating the G at Inr + 2 position of the Inr to T or A at the stably paused promoters *dve*, *pk* and the synthetic Super Core Promoter (SCP). (C) Pol II ChIP-nexus profiles after treatment with triptolide (TRI) for 1 hr show a strong reduction in Pol II pausing after the G was mutated at the *dve* promoter. (D) Quantification of the relative changes in paused Pol II stability for all fusion promoters relative to the corresponding wild-type promoter. Error bars are from replicate experiments. The duration of TRI treatment was 1 hr for *pk* and *dve* derived promoters, 30 min for SCP derived promoters.
DOI: https://doi.org/10.7554/eLife.41461.017

The following figure supplements are available for figure 7:

**Figure supplement 1.** Correlation between paused Pol II half-life and Inr variants using a non-mutually exclusive model.
DOI: https://doi.org/10.7554/eLife.41461.018

**Figure supplement 2.** Mutating the G at Inr + 2 position reduces Pol II pausing at SCP and *pk*.
DOI: https://doi.org/10.7554/eLife.41461.019

**Figure supplement 3.** Transcript level changes after altering the promoter sequence.
DOI: https://doi.org/10.7554/eLife.41461.020

overall transcription levels (*Figure 7—figure supplement 3*), consistent with in vitro studies suggesting that the G is not important for transcription efficiency (*Lo and Smale, 1996*).

Strikingly, we observed a dramatic reduction in the stability of paused Pol II in the mutated constructs relative to the wild-type construct (*Figure 7C and D*). The *dve* promoter showed an almost five-fold decrease when mutated to either T or A. Even the *pk* and SCP promoters, which have more downstream pausing elements (Table S1) showed a ~ 2 fold reduction. These results show that the G in the Inr is indeed critical for stable Pol II pausing in *Drosophila*. More generally, this discovery demonstrates that reporter ChIP-nexus can be used to analyze the role of promoter sequences in Pol II pausing and uncover previously unknown roles of promoter elements.

## Discussion

### The Inr plays an important role in stable Pol II pausing

Genome-wide correlations have suggested a role for promoter sequences in Pol II pausing (*Chen et al., 2013*; *Gaertner et al., 2012*; *Gilchrist et al., 2010*; *Hendrix et al., 2008*; *Nechaev et al., 2010*; *Shao and Zeitlinger, 2017*), but functional data supporting a causal role for these sequences had been largely lacking. Here, we found that Pol II pausing is remarkably well recapitulated on a plasmid and that its stability can be measured on the plasmid by treatment with triptolide. By taking advantage of this assay, which we call reporter-ChIP-nexus, we were able to test the contribution of individual core promoter elements to Pol II pausing. We found that replacement of core promoter regions bound by TFIID tended to change the stability of paused Pol II in the expected direction, but the overall sequence context of the promoter was still important.

Most notably, we found that the Inr sequence played a strong role in the stability of Pol II pausing. This is surprising because the Inr sequence is thought to function synergistically with both the TATA box and the pausing elements (*Juven-Gershon and Kadonaga, 2010*), which have opposite effects on the stability of Pol II pausing. Identifying the preferred Inr sequences in promoters with a TATA box and with pausing elements resolved this conundrum. Promoters with stable Pol II pausing are highly enriched for Inr sequences resembling the consensus TYAGTY, which closely matches the Inr derived from functional studies (TCAKTY) (*Hultmark et al., 1986*; *Purnell et al., 1994*) and from computational analyses (TCAGTY and TCAGTT) (*FitzGerald et al., 2006*; *Ohler et al., 2002*; *Stark et al., 2007*). This in turn suggests that a strong consensus Inr sequence promotes Pol II pausing.

Promoters with a TATA box, in contrast, tend to have more degenerate Inr sequences. This explains why previous genomic sequence analyses have found these two elements to co-occur relatively infrequently in both *Drosophila* and human promoters (*Chen et al., 2013*; *FitzGerald et al., 2006*; *Jin et al., 2006*; *Ohler et al., 2002*; *Vo Ngoc et al., 2017a*), although biochemical experiments clearly show that they function synergistically in transcription (*Emami et al., 1997*; *Malecová et al., 2007*; *O'Shea-Greenfield and Smale, 1992*; *Xu et al., 2011*).

This difference in Inr sequence between promoter types is best reflected in the presence or absence of a G at the +2 position. Although G and T at this position are equally functional in transcription assays in vitro (*Lo and Smale, 1996*), stably paused promoters predominantly contain the Inr-G variant. Mutating this G to an A or T drastically reduced Pol II pausing in our assay, suggesting that the G is indeed important for stable Pol II pausing.

It is therefore possible that the degenerate Inr-nonG sequences in TATA-containing promoters represent a tradeoff. On one hand, these Inr sequences probably have sufficient sequence information to influence start site selection and promote TATA-dependent transcription. On the other hand, they may be weakened in their ability to stabilize Pol II pausing and thereby allow a mode of transcription with reduced Pol II pausing. Alternatively, TATA-containing promoters may function equally well with or without Pol II pausing, and therefore, the Inr sequences in these promoters may not have been under strong purifying selection during evolution.

Interestingly, an Inr-G variant (TCAGTT) was identified in comparative *Drosophila* genomics analyses as the most conserved Inr sequence (*Stark et al., 2007*). This suggests that the G variant may be under evolutionarily selection in paused promoters, consistent with Pol II pausing playing a critical role in keeping the promoter open (*Gilchrist et al., 2008*) and achieving coordinated gene expression during development (*Lagha et al., 2013*).

It is important to point out that while the G is conserved in *Drosophila*, human Inr sequences are often more degenerate and do not show a significant enrichment of G at the +2 (*Carninci et al., 2006*; *Frith et al., 2008*; *Lo and Smale, 1996*; *Vo Ngoc et al., 2017a*). However, there are also Inr variants with a much longer consensus sequence (*Hendy et al., 2017*; *Yarden et al., 2009*). It is therefore possible that human Inr variants also differentially affect Pol II pausing.

### Promoter elements could be linked to Pol II pausing through TFIID

The promoter elements we tested here for Pol II pausing have traditionally been studied for their role in promoter recognition and transcription initiation by TFIID (*Burley and Roeder, 1996*; *Lee and Young, 2000*). TFIID is a large, flexible multi-subunit complex that can simultaneously bind

to the TATA box, the Inr, and the pausing elements (*Louder et al., 2016*; *Patel et al., 2018*), thereby promoting the assembly of the transcription initiation complex. More recent evidence, however, suggests that TFIID has a function beyond initiation. For example, there is accumulating evidence that TFIID promotes Pol II re-initiation (*Joo et al., 2017*; *Oelgeschläger et al., 1998*; *Yudkovsky et al., 2000*; *Zhang et al., 2015*). Furthermore, genome-wide ChIP-nexus data show that TFIID binding extends beyond the pausing position in vivo (*Shao and Zeitlinger, 2017*), raising the intriguing possibility that TFIID plays a role in the coordination between consecutive rounds of Pol II initiation, pausing, and pause release.

If TFIID remains bound after initiation, it is possible that it influences the stability of Pol II pausing in a promoter-specific fashion. While TFIID can make contacts with all promoter elements on the super core promoter (*Louder et al., 2016*; *Patel et al., 2018*), natural promoters contain fewer promoter element consensus sequences, such that the strength and position of TFIID-promoter contacts may vary between promoters. We therefore speculate that the core promoter elements recognized by TFIID contribute to the dynamics of Pol II transcription by influencing how Pol II enters and remains in the pausing position (*Figure 8*). If TFIID binds tightly to the Inr and downstream promoter elements, Pol II may initiate and traverse the early transcribed region more slowly, and hence may be more likely to become stably paused (*Figure 8A*). In contrast, if TFIID is more tightly bound to the promoter upstream through the TATA box, then Pol II initiation and early transcription may occur in a less constrained fashion, which may promote pause release (*Figure 8B*). These tight DNA contacts in the presence of the TATA box is supported by biochemical experiments (*Lee et al., 1991*; *Starr and Hawley, 1991*), as well as more recent single-molecule footprinting studies (*Krebs et al., 2017*). In summary, although all core promoter elements may help recruit TFIID, they may differentially affect the passage of Pol II and its release into productive elongation, dependent on their location at the promoter.

## Future perspective

We found that reporter-ChIP-nexus is a useful method for studying the relationship between promoter sequence and Pol II pausing. This plasmid-based assay recapitulates the endogenous Pol II dynamics at the promoters tested and has sufficient sensitivity to identify changes in Pol II pausing upon promoter sequence alterations. The simplicity of the assay will no doubt be helpful for further studies on the roles of individual promoter sequences in Pol II pausing, including those of promoter types not studied here.

In order to perform a more high-throughput analysis on promoter sequences, some improvements in the assay will be beneficial. First, we currently sequence Pol II ChIP-nexus reads genome-wide in order to capture the reads from the plasmid. Any prior enrichment for plasmid sequences will therefore lower the cost of sequencing. Second, if genomic reads can no longer be reliably used

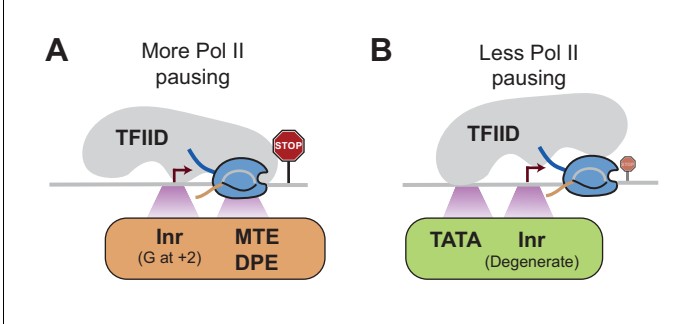

**Figure 8.** Model of how promoter sequences may influence Pol II pausing through TFIID Since the core promoter elements that we analyzed are bound by TFIID, we hypothesize that TFIID affects Pol II pausing dependent on the position it binds to the promoter. (A)At stably paused promoters, pausing elements and consensus Inr sequences with a G at +2 position promote TFIID-downstream DNA interactions, which prolong Pol II pausing. (B)At TATA-containing promoters, the TATA box and a degenerate Inr sequence favor TFIID interactions with upstream DNA, which promote Pol II pause release.
DOI: https://doi.org/10.7554/eLife.41461.021

as internal control, a control plasmid that monitors transfection efficiency would be useful. Finally, the assay will greatly benefit from automation. This would not only enable more high-throughput analyses but also improve the scope of the assay. Currently, the accuracy and reproducibility of the assay is higher for promoters with longer Pol II half-lives since longer triptolide treatment times reduce experimental fluctuations in handling and drug penetration. If automated, the assay would allow more tightly spaced time-course experiment in multiple replicates, which could more accurately distinguish smaller differences in Pol II half-lives.

Taken together, reporter-ChIP-nexus promises to open new possibilities for measuring the stability of Pol II pausing at individual promoter sequences in the future. Together with complementary approaches such as live imaging analyses, such studies will contribute to a better understanding of the relationship between promoter sequence, Pol II pausing, and the dynamic production of transcripts over time.

# Materials and methods

## Key resources table

| Reagent type (species) or resource | Designation | Source or reference | Identifiers | Additional information |
|---|---|---|---|---|
| Antibody | Rabbit polyclonal anti-Rpb3 | Julia Zeitlinger Lab | Zeitlinger Lab #163185–50 | 10 ug |
| Antibody | Rabbit polyclonal anti-H3K4me3 | Cell Signaling | #9727 | 10 ug |
| Strain, strain background (E. coli) | NEB 5-alpha Competent E. coli (High Efficiency) | New England Biolabs | #C2987H | |
| Strain, strain background (E. coli) | One Shot ccdB Survival 2 T1R Competent Cells | ThermoFisher Scientific | #A10460 | |
| Chemical | Chloroform | Sigma-Aldrich | #C2432 | |
| Commercial assay or kit | CircLigase ssDNA Ligase | Illumina (Epicentre) | #CL4115K | |
| Chemical compound | DMSO | Sigma-Aldrich | #276855 | |
| Chemical compound | Dynabeads Protein A | Life Technologies | #100-08D | |
| Chemical compound | Dynabeads Protein G | Life Technologies | #100-04D | |
| Commercial assay or kit | FastDigest BamHI | ThermoFisher Scientific | #FERFD0054 | |
| Chemical compound | FuGENE HD reagent | Promega | #E2311 | |
| Chemical compound | HyClone SFX-Insect Cell Culture Media | ThermoFisher Scientific | #SH3027802PM | |
| Commercial assay or kit | Klenow Fragment (3′ to 5′ exo-) | New England Biolabs | #M0212S | |
| Commercial assay or kit | Lambda Exonuclease | New England Biolabs | #M0262L | |
| Chemical compound | Opti-MEM Reduced Serum Medium | ThermoFisher Scientific | #31985062 | |
| Chemical compound | Phenol: Chloroform: iso-Amyl alcohol | VWR | #97064–824 | |
| Chemical compound | Protease Inhibitor Cocktail tablets EDTA-free | Roche Diagnostics Corporation | #5056489001 | |

*Continued on next page*

*Continued*

| Reagent type (species) or resource | Designation | Source or reference | Identifiers | Additional information |
|---|---|---|---|---|
| Commercial assay or kit | Proteinase K | Life Technologies | #25530–049 | |
| Commercial assay or kit | recJ | New England Biolabs | #M0264L | |
| Commercial assay or kit | Restriction emzyme AfeI | New England Biolabs | #R0652S | |
| Commercial assay or kit | Restriction emzyme EcoRV-HF | New England Biolabs | #R3195S | |
| Commercial assay or kit | Restriction emzyme SacI-HF | New England Biolabs | #R3156S | |
| Commercial assay or kit | RNase A | ThermoFisher Scientific | #EN0531 | |
| Commercial assay or kit | RNase H | New England Biolabs | #M0297S | |
| Commercial assay or kit | SuperScript II Reverse Transcriptase | ThermoFisher Scientific | #18064014 | |
| Commercial assay or kit | T4 DNA Polymerase | New England Biolabs | #M0203S | |
| Chemical compound | Triptolide | TOCRIS Bioscience | #3253 | |
| Chemical compound | TRIzol Reagent | ThermoFisher Scientific | #15596026 | |
| Commercial assay or kit | Direct-zol RNA MiniPrep kit | Genesee | #11–330 | |
| Commercial assay or kit | Fast SYBR Green Master mix | Life Technologies | #4385612 | |
| Commercial assay or kit | Gibson Assembly Master Mix | New England Biolabs | #E2611S | |
| Commercial assay or kit | High-Capacity RNA-to-cDNA Kit | ThermoFisher Scientific | #4387406 | |
| Commercial assay or kit | IBI High Speed Plasmid Mini Kit | MIDSCI | #IB47101 | |
| Commercial assay or kit | NEBNext dA-tailing module | New England Biolabs | #E6053L | |
| Commercial assay or kit | NEBNext end-repair module | New England Biolabs | #E6050L | |
| Commercial assay or kit | NEBNext Multiplex Oligos for Illumina | New England Biolabs | #E7335S | |
| Commercial assay or kit | Q5High-Fidelity 2X Master Mix | New England Biolabs | #M0492L | |
| Commercial assay or kit | Q5 Site-Directed Mutagenesis Kit | New England Biolabs | #E0554S | |
| Commercial assay or kit | Quick Ligation Kit | New England Biolabs | #M2200L | |
| Cell line (*D. melanogaster*) | Kc167 cells | DGRC | #1 | |
| Cell line (*D. pseudoobscura*) | ML83-63 cells | DGRC | #33 | |

*Continued on next page*

*Continued*

| Reagent type (species) or resource | Designation | Source or reference | Identifiers | Additional information |
|---|---|---|---|---|
| Sequence-based reagent | ChIP-nexus oligos | (*He et al., 2015*) | | See Table S5 |
| Sequence-based reagent | Gene-specific 5' RNA sequencing oligos | This paper | | See Table S6 |
| Recombinant DNA reagent | Reporters used in this study | This paper | | See Table S7 and S8 |
| Software, algorithm | Analysis code | GitHub | https://github.com/zeitlingerlab/Shao_eLife_2019 | |

## Methods

### Reporter construction

The pAWG GFP reporter plasmid from the *Drosophila* Gateway cloning collection was used as the backbone for the reporter plasmid. The *Act5C* core promoter (−41 to 103 bp around the transcription start site, not including the upstream Act5C regulatory sequences, which is −2470 to −42 bp from the transcription start site) and the downstream Gateway cloning cassette (sequence between attR1 and attR2) were removed by digesting with the SacI and AfeI restriction enzymes. A 6 x UAS sequence from pUASp (as found in DGRC #1189) was inserted between the SacI and AfeI restriction sites using Gibson Assembly Master Mix. The resulting plasmid pAWG-UAS has the *Act5c* upstream regulatory region, 6 x UAS, a GFP coding sequence and an EcoRV restriction site between the UAS sequences and the GFP coding region. Promoter sequences of interests were inserted into pAWG-UAS at the EcoRV cutting site with Gibson Assembly Master Mix. Subsequent mutations of core promoter sequences were performed with the Q5 site-directed site mutagenesis kit. For plasmid amplification and purification, the DNA was transformed into NEB 5-alpha Competent E. coli (High Efficiency) or One Shot ccdB Survival 2 T1R Competent Cells and purified with IBI High Speed Plasmid Mini Kit. Clones were validated with Sanger sequencing using the sequencing primer GCACCG TGACCATCACAGCATA. See Table S6 and S7 for detailed constructs descriptions and sequences.

### Cell culture and transcription inhibitor treatment

*D. melanogaster* Kc167 cells (DGRC #1) were grown in SFX media at 25°C. *D. pseudoobscura* ML83-63 cells (DGRC #33) were grown in M3 +BPYE + 10% FCS media at 25°C. Transcription inhibitors were added directly into culture media. Cells were treated with 500 µM Triptolide (TOCRIS Bioscience Cat. No. 3253 dissolved in DMSO) as done previously (*Shao and Zeitlinger, 2017*). Equivalent amounts of DMSO treatment (2% v/v) were used as control. To best capture the stability of paused Pol II, cells were treated with Triptolide for different time after transfecting with different reporter constructs. Refer to the main text and figure legends for the treatment duration.

### ChIP-nexus

For each ChIP-nexus experiment, $10^7$ Kc167 cells or ML83-63 cells were fixed with 1% formaldehyde in culturing media at room temperature for 10 min with rotation. Fixed cells were washed with cold PBS, incubated with Orlando and Paro's Buffer (0.25% triton X-100, 10 mM EDTA, 0.5 mM EGTA, 10 mM Tris-HCl pH 8.0, with freshly added Protease Inhibitor) for 10 min at room temperature with rotation, and then centrifuged and re-suspended in ChIP Buffer (10 mM Tris-HCl, pH 8.0; 140 mM NaCl; 0.1% SDS; 0.1% sodium deoxycholate; 0.5% sarkosyl; 1% Triton X-100, with freshly added Protease Inhibitor). Sonication was performed with a Bioruptor Pico for five rounds of 30 s on and 30 s off. Chromatin extracts were then centrifuged at 16000 g for 5 min at 4°C, and supernatants were used for ChIP.

To couple Dynabeads with antibodies, 50 µl Protein A and 50 µl Protein G Dynabeads were used for each ChIP-nexus experiment and washed twice with ChIP Buffer. After removing all the liquid, Dynabeads were resuspended in 400 µl ChIP Buffer. 10 µg antibodies were added, and tubes were

incubated at 4°C for 2 hr with rotation. After the incubation, antibody-bound beads were washed twice with ChIP Buffer.

For chromatin immunoprecipitation, chromatin extracts were added to the antibody-bound beads and incubated at 4°C overnight with rotation and then washed with Nexus washing buffer A to D (wash buffer A: 10 mM Tris-EDTA, 0.1% Triton X-100, wash buffer B: 150 mM NaCl, 20 mM Tris-HCl, pH 8.0, 5 mM EDTA, 5.2% sucrose, 1.0% Triton X-100, 0.2% SDS, wash buffer C: 250 mM NaCl, 5 mM Tris-HCl, pH 8.0, 25 mM HEPES, 0.5% Triton X-100, 0.05% sodium deoxycholate, 0.5 mM EDTA, wash buffer D: 250 mM LiCl, 0.5% IGEPAL CA-630, 10 mM Tris-HCl, pH 8.0, 0.5% sodium deoxycholate, 10 mM EDTA). End repair and dA-tailing were performed using the NEBNext End Repair Module and the NEBNext dA-Tailing Module. ChIP-nexus adaptors with mixed fixed barcodes (CTGA, TGAC, GACT, ACTG) were ligated with Quick T4 DNA ligase and converted to blunt ends with Klenow fragment and T4 DNA polymerase. The samples were treated with lambda exonuclease and RecJ$_f$ exonuclease for generating Pol II footprints at high resolution. After each enzymatic reaction, the chromatin was washed with the Nexus washing buffers A to D and Tris buffer (10 mM Tris, pH 7.5, 8.0, or 9.5, depending on the next enzymatic step).

After RecJ$_f$ exonuclease digestion, the chromatin was eluted and subjected to reverse crosslinking and ethanol precipitation. Purified single-stranded DNA was then circularized with CircLigase, annealed with oligonucleotides complementary to the BamHI restriction site and linearized by BamHI digestion. The linearized single-stranded DNA was purified by ethanol precipitation and subjected to PCR amplification with NEBNext High-Fidelity 2X PCR Master Mix and ChIP-nexus primers. The ChIP-nexus libraries were then gel-purified before sequencing with Illumina NextSeq 500.

### Reporter-ChIP-nexus

To transform cells with the reporter plasmid, $1.5 \times 10^7$ Kc167 cells were diluted in 10 ml SFX media and seeded into a 10 cm cell culture dish. A mixture of 2.5 µg reporter plasmids, 20 µl FuGENE HD reagent and 500 µl Opti-MEM Reduced Serum Medium was then added to the cell culture. Transfected cells were cultured for 48 hr at 25°C to allow proper expression of the reporter. ChIP-nexus was performed on the whole cell extracts from transfected cells following standard ChIP-nexus procedure as described above. After sequencing, reporter-ChIP-nexus samples were aligned to the combined genome of dm3 and the specific reporter construct. Only reads that uniquely aligned to the reporter sequence were used for analysis and PCR duplicates with the same ChIP-nexus barcode were removed.

### Reporter expression quantification

Expression of the reporter was quantified by calculating the expression of GFP RNA from the reporter over the expression of the endogenous RNA from the *RpL30* locus, after normalizing for the difference in copy number between the reporter and the endogenous genome.

For cDNA preparations, $5 \times 10^5$ transfected cells were transferred into 1.5 ml tubes and washed with cold PBS. After removing all liquid, cells were lysed and incubated with 500 µl TRIzol Reagent at room temperature for 5 min. Total RNA was then extracted with 100 µl Chloroform and purified with a Direct-zol RNA MiniPrep kit. cDNA was generated with a High-Capacity RNA-to-cDNA Kit.

For DNA preparations, $5 \times 10^5$ transfected *Drosophila* Kc167 cells were transferred to 1.5 ml tubes and washed with cold PBS. After removing all liquid, cells were lysed with 50 µl ChIP buffer. Cell lysates were incubated with 150 Elution buffer (50 mM Tris pH 8.0, 10 mM EDTA, 1% SDS), 100 µl TE buffer (50 mM Tris pH 8.0, 1 mM EDTA) and 4 µl RNAse A for 30 min at 37°C. Then, 2 µl Protease K was added to the mixture and incubated at 65°C for 2 hr. The DNA was then purified with an ethanol precipitation. qPCR on both the cDNA and DNA was performed using Fast SYBR Green Master Mix and primers against GFP and *RpL30*. The relative expression of the reporter was calculated using the following equations.

$$\mathrm{GFP\,expression} = 2^{\mathrm{Ct(GFP\,DNA)-Ct(GFP\,cDNA)}}$$

$$\mathit{RpL}30\,\mathrm{expresssion} = 2^{\mathrm{Ct(RpL30\,DNA)-Ct(RpL30\,cDNA)}}$$

$$\mathrm{GFP\,relative\,expression} = (\mathrm{GFP\,expression})\,/\,\mathrm{RpL30\,expression}$$

For this analysis, cDNA was prepared from two to three biological replicates (cells were transfected and processed on different dates) using $5 \times 10^5$ transfected cells each. Two technical replicates (the same biological sample were split into two and processed side by side) were performed during qPCR.

## Gene-specific 5' RNA sequencing

This method is similar to 5' RNA amplification of cDNA ends (RACE) but uses reagents and steps from ChIP-nexus. Briefly, total RNA was isolated from $5 \times 10^5$ transfected cells as above, and cDNA was generated from the GFP RNA using SuperScript II Reverse Transcriptase and a primer against GFP that has partial TruSeq P5 sequences at the 5' end (GFP_reversetx_primer). The cDNA was purified by treatment with RNAse A and RNAse H at room temperature for 30 min, followed by ethanol precipitation. The cDNA circularization, cutting and PCR amplification was then performed as in the ChIP-nexus protocol (thus without rolling circle amplifications), except that a new BamHI cutting oligo (reversetx_BamHI_cutting) was annealed before BamHI cutting, and a barcoded P7-GFP fusion primer (reversetx_barcode_primer_#1) was used in the PCR. After sequencing with TruSeq primers, the reads were aligned to the dm3-reporter combined genome, and the first bases of the P5 sequenced reads were recorded as the stop bases of the reverse transcriptase and correspond to the 5' ends of the GFP RNA from the reporter plasmid.

## Quantification and statistical analysis

### Normalization of Pol II signal on the plasmid

Due to variations in the transfection efficiency, absolute Pol II signal on different plasmids is not comparable. As a result, paused Pol II stability needs to be accessed by comparing Pol II signal under control and triptolide-treated conditions. Because control and triptolide-treated samples come from the same pool of transfected cells, they have the same transfection efficiency and the same plasmid/Kc167 genome ratio. Therefore, Pol II signal on the plasmids under control and triptolide-treated conditions can be directly compared after normalizing for read counts.

The changes in Pol II signal after triptolide treatment is calculated in the following way:

1. Each sample is first normalized to reads/millions to account for differences in read coverage.
2. For each sample, the total Pol II signal is calculated in a 301 bp window around the promoter region on the plasmid (termed Total_Pol_sig).
3. The change in Pol II signal on the plasmid in response to triptolide treatment is then calculated as the ratio between Total_Pol_sig from the triptolide-treated sample over Total_Pol_sig from the corresponding DMSO-treated control sample.

It is likely that this method underestimates the changes of Pol II signal between control and triptolide-treated condition since the total occupancy of Pol II is slightly reduced after triptolide treatment (*Shao and Zeitlinger, 2017*), yet we normalize both samples to equal read counts. However, since the relative loss of Pol II after triptolide treatment is similar across all samples in the Kc167 genome, the relative differences in the above calculated ratio between different plasmids is preserved. For this analysis, two biological replicates (cells were transfected and processed on different dates) were performed using $10^7$ transfected cells each.

### Correlation between paused Pol II half-lives and core promoter elements

In *Figure 4A* and *Figure 4—figure supplement 1*, promoters with previously measured paused Pol II half-life (*Shao and Zeitlinger, 2017*) were separated into groups with either one or two of the following core promoter elements: TATA box (STATAWAWR), Inr (TCAKTY) and pausing elements (CSARCSSA, KCGGTTSK or KCGRWCG). Only motif matches at the expected canonical position and with up to one mismatch were allowed (Table S1). In a mutually exclusive model (*Figure 4—figure supplement 1*), promoters with TATA-box and Inr cannot contain pausing elements, while in a non-mutually exclusive model (*Figure 4A*), pausing elements are allowed.

The same method was applied to *Figure 7A,B* (using a mutually exclusive model) and *Figure 7—figure supplement 1* (using a non-mutually exclusive model), except that promoters were further separated into containing Inr-G or Inr-nonG variants based on whether a G is present at the +2 position.

## Sequence analysis of TATA promoters and stably paused promoters

TATA promoters and stably paused promoters were defined using the length of paused Pol II half-life (*Shao and Zeitlinger, 2017*) and the presence or absence of the TATA box sequence (STATA-WAWR with up to one mismatch) at 40 to 20 bp upstream of the transcription start site. TATA promoters have the TATA box sequence and paused Pol II half-lives shorter than 30 min, whereas stably paused promoters lack a detectable TATA box sequence and have paused Pol II half-lives longer than 60 min. In *Drosophila* Kc167 cell, 132 promoters were defined as TATA promoters and 490 promoters were defined as stably paused promoters.

In *Figure 5A*, DNA sequences at TATA promoters (n = 132) and stably paused promoters (n = 132, randomly selected from the original 490 promoters) were obtained from the dm3 genome and represented as heatmap. The consensus motif for the Inr sequence in *Figure 5B* was generated using the R package seqLogo.

### Data and software availability

Raw and processed data associated with this manuscript are deposited in GEO under the accession number GSE116244 (https://www.ncbi.nlm.nih.gov/geo/query/acc.cgi?acc=GSE116244).

All data analysis performed in this paper, including raw data, processed data, software tools, and analysis scripts, have been reproduced in a publicly accessible Linux virtual machine. Instructions for accessing the virtual machine can be found at http://research.stowers.org/zeitlingerlab/data.html. The analysis code is available on GitHub at https://github.com/zeitlingerlab/Shao_eLife_2019 (*Shao, 2019* ; copy archived at https://github.com/elifesciences-publications/Shao_eLife_2019).

## Acknowledgements

We thank our colleagues Joan Conaway, Robb Krumlauf and Kaelan Brennan at the Stowers Institute for Medical Research, Mounia Lagha (Institut de Génétique Moléculaire de Montpellier) and James Kadonaga (UC San Diego) for comments on the manuscript. The work was funded by the Stowers Institute for Medical Research and has been done to fulfill, in part, requirements for Wanqing Shao's PhD thesis research as a student of the Graduate School of Stowers Institute for Medical Research and for Sergio Garcia-Moreno's PhD thesis research as a student registered with the Open University.

## Additional information

### Competing interests

Julia Zeitlinger: J.Z. owns a patent on ChIP-nexus. EP3234199A1. The other authors declare that no competing interests exist.

### Funding

| Funder | Author |
| --- | --- |
| Stowers Institute for Medical Research | Julia Zeitlinger |

The funders had no role in study design, data collection and interpretation, or the decision to submit the work for publication.

### Author contributions

Wanqing Shao, Conceptualization, Software, Formal analysis, Validation, Investigation, Visualization, Methodology, Writing—original draft, Writing—review and editing.; Sergio G-M Alcantara, Conceptualization, Validation, Investigation, Methodology, Writing—review and editing.; Julia Zeitlinger, Conceptualization, Data curation, Formal analysis, Supervision, Funding acquisition, Validation, Investigation, Visualization, Methodology, Writing—original draft, Project administration, Writing—review and editing.

## Author ORCIDs

Wanqing Shao ![ORCID] https://orcid.org/0000-0003-4234-2095
Sergio G-M Alcantara ![ORCID] https://orcid.org/0000-0003-3695-4677
Julia Zeitlinger ![ORCID] https://orcid.org/0000-0002-5172-3335

## Decision letter and Author response

Decision letter https://doi.org/10.7554/eLife.41461.029
Author response https://doi.org/10.7554/eLife.41461.030

## Additional files

### Supplementary files

• Supplementary file 1. 2019 eLife Shao Manuscript Supplementary Information.
DOI: https://doi.org/10.7554/eLife.41461.022

• Transparent reporting form
DOI: https://doi.org/10.7554/eLife.41461.023

### Data availability

Raw and processed data associated with this manuscript are deposited in GEO under session number GSE116244. All data analysis performed in this paper, including raw data, processed data, software tools, and analysis scripts, have been reproduced in a publicly accessible Linux virtual machine. Instructions for accessing the virtual machine can be found at http://research.stowers.org/zeitlinger-lab/data.html. The analysis code is available on GitHub at https://github.com/zeitlingerlab/Shao_eLife_2019 (copy archived at https://github.com/elifesciences-publications/Shao_eLife_2019).

The following dataset was generated:

| Author(s) | Year | Dataset title | Dataset URL | Database and Identifier |
|---|---|---|---|---|
| Shao W, Alcantara SG-M, Zeitlinger J | 2019 | Reporter-ChIP-nexus reveals strong contribution of the *Drosophila* initiator sequence to RNA polymerase pausing | https://www.ncbi.nlm.nih.gov/geo/query/acc.cgi?acc=GSE116244 | NCBI Gene Expression Omnibus, GSE116244 |

The following previously published dataset was used:

| Author(s) | Year | Dataset title | Dataset URL | Database and Identifier |
|---|---|---|---|---|
| Shao W, Zeitlinger J | 2017 | Paused RNA polymerase II inhibits new transcriptional initiation | https://www.ncbi.nlm.nih.gov/geo/query/acc.cgi?acc=GSE85741 | NCBI Gene Expression Omnibus, GSE85741 |

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
