## [Decision Letter]

Thank you for sending your article entitled "Reporter-ChIP-nexus reveals contribution of initiator sequence to RNA Polymerase II pausing" for peer review at *eLife*. Your article has been evaluated by three peer reviewers, one of whom is a member of our Board of Reviewing Editors, and the evaluation has been overseen by Aviv Regev as the Senior Editor.

Summary:

In this work, the Zeitlinger lab uses ChIP-nexus (a high-resolution form of ChIP-seq developed in the Zeitlinger lab) combined with a reporter system to study the contribution of promoter DNA sequence to the stability of paused polymerase. Using *D. pseudoobscura* promoters in the reporter plasmid in a *D. melanogaster* genomic background, they show that initiation, pausing, and the stability of pausing are recapitulated on the plasmids. This is important validation of the usefulness of the system. Promoter swapping and targeted mutagenesis of reporter promoters then yielded three main results: 1) the downstream region of promoters was important for pause stability, 2) a TATA-box is associated with pause instability, and 3) a strong initiator consensus with a G at +2 was critical for pause stability.

The reviewers all found the ChIP-nexus technique to be powerful, with the potential for high throughput screens with this strategy. However, as presented here, the assay is very low throughput. Incredibly, most of the authors conclusions are drawn from a single perturbation of single test promoter sequence. The work herein is thus considered preliminary, and requires additional validation prior to publication.

In short, the very limited, highly context-dependent data presented in this manuscript reduces enthusiasm for both the assay and the strength of the authors' conclusions. If concerns can be addressed with additional experiments, however, the reviewers are happy to consider a revised manuscript.

Essential revisions:

1) The authors must test each of the promoter mutations in several promoter contexts to validate findings. We recommend that every mutation be tested in at least 3 different promoter backgrounds to allow for reasonable confidence in conclusions.

2) The drug triptolide has been used by a number of groups to measure the stability of paused RNAPII and all data thus far converges towards a median half-life of paused RNAPII on the order of around 5-10 min. Genes that have polymerase remaining at 1 h of triptolide treatment are thus extreme outliers. As such, it is a shame that the authors have selected this type of hyper-stable promoter for use in their assay, as it makes it very difficult- if not impossible- to extrapolate from this extreme case to normal promoter behavior. In the absence of experiments on a more typical promoter, the reviewers are concerned that the results presented here will not be broadly applicable. Thus, when additional promoters are tested, we strongly recommend that the authors select promoters with decay rates more in line with the average gene, and use a more relevant time point for triptolide treatment, on the order of 5-20 min.

3) To better support the authors claims, they should perform a genomic analysis with their existing data on the stability of pausing at promoters with various combinations of the TATA and Inr elements. They already have the *melanogaster* data from Kc cells +/- triptolide and should have plenty of sequencing depth given the number of plasmids analyzed. Thus, in theory, this would simply require analysis of the genomic reads without additional experimentation. This would serve to corroborate the results and generalize the model to more promoters.

4) The claim that the reporter assay recapitulates the genomic assay could also use strengthening. This point is supported by 8 genes in the current manuscript, which is probably not enough for statistically rigorous conclusions. Figure 2—figure supplement 2 shows the 8 examples, and there is quite some variability in them. The question is how much variability, how to quantify it, and whether or not conclusions can be made given this variability. Given that a lot of conclusions are drawn from comparing the shapes of these promoter distributions, I would think the extent to which they can be reliably made is important. dve, the gene selected for the majority of mutational tests, is an example that shows differences between genomic and reporter contexts.

One idea is to bin the promoters into, say 10-20bp bins, and compare the normalized read counts across the bins for the reporter and genomic assay. Then the mean squared error or the correlation coefficient can be computed comparing the reporter and the genomic assay, and the degree of variability can be measured across more genes. This would at least give us some context for how much variability can be expected, and if the differences observed due to the mutations are significantly more than that baseline difference. This analysis or something similar is considered critical.

5) The novelty of finding a strong initiator at highly paused genes is severely over-exaggerated and previous literature is inadequately cited. The connection between Initiator consensus elements, highly focused transcription initiation and high levels of paused RNAPII has appeared multiple times in the literature previously, at least as far back as 2008 (Hendrix et al., 2008), and also in 2010 (Gilchrist et al., 2010).

Further, pausing has been recapitulated on DNA sequences in vitro, thus we already know that DNA sequence is a major contributor to pausing. See many in vitro studies from the Gilmour lab (some cited) and also Adelman and Lis (2005).

Finally, the authors mention in the Discussion that functional data supporting a role for promoter sequences and pausing are missing. While they are correct that there is a dearth of these important functional studies, the Lis lab did a functional analysis pause elements in the *Drosophila* Hsp70 promoter, by inserting sequences that changed the position of these elements. See Kwak et al., 2013. Given that there are so few studies like this and the level of detail covering other aspects of the literature, I think this experiment is worth discussing in this context.

It doesn't take anything away from the authors' accomplishments to adequately cite previous research in a scholarly way. Appropriate citations must be added.

6) Also related to the ambiguity in the number of promoters involved in each analysis: In the section "The Inr strongly contributes to the degree of Pol II pausing", I think it would be enhanced if the text listed the number of promoters examined for each group. For example, the text says "We first analyzed the naturally occurring Inr sequences from TATA-containing promoters versus those of stably paused promoters". I would like to know when reading this how many promoters are in each set leading to the observations e.g. that TATA-containing promoters have fewer Gs at the +2 position. In particular, when reading the caption for Figure 5A, it says n=132, and this implied that the other panels also used 132 promoters. However I expect that the other panels were based on a larger set (at least panel B). Ideally, more than 132 promoters would have been used for this analysis. Overall, added clarity on the number of promoter for Figure 5B and elsewhere would be helpful.

[Editors' note: further revisions were requested prior to acceptance, as described below.]

Thank you for submitting your article "Reporter-ChIP-nexus reveals strong contribution of initiator sequence to RNA Polymerase II pausing" for consideration by *eLife*. This evaluation has been overseen by a Reviewing Editor and James Manley as the Senior Editor.

The reviewers have discussed the reviews with one another and the Reviewing Editor has drafted this decision to help you prepare a revised submission.

Summary:

This revised version of the manuscript "Reporter-ChIP-nexus reveals strong contribution of initiator sequence to RNA Polymerase II pausing" is improved over the previous version as concerns the number of promoters and mutations analyzed. However, the way in which the data is discussed is still lacking in clarity, the caveats to interpretation are still not made clear, and the citation of prior literature is still lacking.

Thus, the manuscript is not determined to be acceptable in its current format. However, at this point the main criticisms of the work can all be addressed with text changes. Thus, we invite a resubmission if such changes are comprehensively made.

Essential revisions:

1) As the authors acknowledge in the Introduction, pausing takes place at all (or nearly all) genes. The duration and stability of pausing varies, but pausing appears to be an obligate part of the transcription cycle. Thus, the statement in the Abstract that "a G at the +2 position, is critical for Pol II pausing", is not accurate. The authors should change this to clarify that they mean 'stable pausing' or something of this sort. But a G and +2 is certainly not critical for the fundamental process of pausing itself.

2) The authors should note early in the Results section that 1h Triptolide treatment is a very long treatment, and that secondary effects are likely to occur during such extended intervals of transcription inhibition. Most labs use 2-20 minutes of Triptolide treatment to avoid such potential artifacts. We appreciate that the authors must rely on very long treatments to overcome the considerable noise in their assay, but this should be acknowledged as a potential source of artifact, and as a potential limitation to this study. This idea is briefly mentioned in the Discussion, but it must be noted in the Results section to make readers aware of this caveat.

3) The Bentley lab has recently questioned the interpretation of Triptolide data (in particular that using ChIP-based strategies such as in Shao et al. 17), noting that at some genes the Triptolide-mediated block to initiation traps Pol II at the TSS. This effect is evident at the Pino and pepck genes in this study. The authors should cite the recent work from the Bentley lab, and explain whether their ChIP-nexus assay gives them the spatial resolution needed to separate paused from un-initiated Pol II. This will help place the current work in a proper context, and will be clarifying for the field.

4) Based on the limited number of experiments aimed at testing the role of downstream promoter sequences, and the limitations in interpreting such experiments, the authors should be more circumspect in the conclusions drawn from these studies. We find it to be an over-reach that the authors conclude that downstream sequences affect pausing less than expected. Given the lack of data to this point, it is safest to note that this wasn't rigorously evaluated, and leave it at that.

5) We understand that the authors went into this study expecting the TATA box to elicit a strong effect on pausing, and that they were surprised that this is not the case. However, the large amount of inconclusive data presented on this topic, and the discussion thereof, is confusing. If the main conclusion is that TATA doesn't seem to have a strong effect on pause stability, and that any effect is highly context dependent and easily over-ridden by other sequences, then there is a much more straightforward way to present this. We strongly suggest that the authors simplify the presentation of the TATA data to focus on the main conclusion, rather than enumerating all the inconclusive or contradictory findings they obtained.

6) Given that computational assays have routinely defined the Inr consensus to have a G at +2, should this really be called the Inr-G variant? Instead, we recommend that this be considered the consensus. It should be clarified that this is the 'norm' and not something that deviates from the norm.

7) The discussion of pausing in its genomic context versus on a plasmid at the beginning of the Discussion is difficult to understand in light of the literature.

First, there have been a number of studies demonstrating that stable pausing does not involve downstream nucleosomes. The clearest of these demonstrations was Li and Gilmour, EMBO J, 2103). This paper should be cited and the authors should edit the first paragraph to reflect this work, and the fact that it is not at all surprising that pausing could take place on a plasmid or outside of the chromatin context.

8) It was demonstrated many years ago that pausing is rarely inhibitory in its endogenous context, because of the positive effect of the paused polymerase on maintaining open chromatin (Gilchrist G and D 2008). In this study, it was specifically demonstrated that the positive effect of pausing was clear in the endogenous context, but did not occur on a plasmid. Thus, it has been previously established that the effect of pausing on gene output is different on a plasmid than in the genome, and that any minor inhibitory effect seen on a plasmid is NOT borne out in the endogenous locus. Thus, it is perplexing that the authors claim that pausing is inhibitory based on their reporter assay- since this is known to be an inaccurate read-out of endogenous activity. Thus, the comments in the Discussion suggesting that the +2 G version of Inr would be inhibitory to transcription because it promotes pausing are unfounded and should be removed.

---

## [Author Response]

Essential revisions:1) The authors must test each of the promoter mutations in several promoter contexts to validate findings. We recommend that every mutation be tested in at least 3 different promoter backgrounds to allow for reasonable confidence in conclusions.

We agree and have now tested 9 additional promoters (2 for downstream pausing elements, 4 for upstream TATA regions, 2 for Inr mutations, 1 ribosomal gene promoter). In addition to these experiments, we have dissected the role of various promoter combinations computationally in more detail and found that these data strengthen and refine our conclusions.

As expected, we found that the downstream pausing elements stabilize pausing and that addition of a TATA box may reduce pausing. Interestingly though, the effect of TATA is not as strong and consistent as expected, and this can be explained by the strong effect of the Inr sequence (with a G at position +2). Mutating the G at the Inr on the other hand strongly reduces Pol II pausing. Consistent with such strong role in pausing, the type of Inr variant has the strongest correlation with the half-life of paused Pol II among all analyzed elements, an analysis we now added. Therefore, we have not only added additional experiments to support our conclusions, but we have strengthened our finding that the Inr plays an important role in Pol II pausing.

2) The drug triptolide has been used by a number of groups to measure the stability of paused RNAPII and all data thus far converges towards a median half-life of paused RNAPII on the order of around 5-10 min. Genes that have polymerase remaining at 1 h of triptolide treatment are thus extreme outliers. As such, it is a shame that the authors have selected this type of hyper-stable promoter for use in their assay, as it makes it very difficult- if not impossible- to extrapolate from this extreme case to normal promoter behavior. In the absence of experiments on a more typical promoter, the reviewers are concerned that the results presented here will not be broadly applicable. Thus, when additional promoters are tested, we strongly recommend that the authors select promoters with decay rates more in line with the average gene, and use a more relevant time point for triptolide treatment, on the order of 5-20 min.

Apart from the stably paused promoters, we have tested the Pol II profile on plasmids using promoters with a half-life in *D. mel*. of 5 (*Act5C*), 10 (*pino*) and 10 (*pepck*) min, respectively, which are in line with “average” promoters. However, in order to detect significant differences in half-lives between hybrid promoters, we have to use the more stable paused promoters. This is because the accuracy and reproducibility of the assay is higher for promoters with longer half-lives. For promoters with shorter half-lives, we have to use shorter drug treatment times. As a result, small fluctuations in drug treatment time and drug penetration can produce considerable variability in the results. To accurately measure a half-life in the order of 5-20 min, we would need to perform tightly spaced time-course experiment in many replicates. In the future, this might be possible as we plan to automate the assay and reduce sequencing cost by isolating plasmid DNA. However, with the current low-throughput method, using sequences from stably paused promoters is a more feasible approach for obtaining clear and interpretable results.

In fact, this is one of the reasons we pioneered this assay in *Drosophila. Drosophila* has a much larger fraction of promoters with clearly defined core promoter sequences that are associated with stable Pol II pausing. While pausing may not be as stable in mammalian genomes, the role of core promoter elements and the interaction between core promoter elements and TFIID is nevertheless conserved. For example, the excellent Cryo-EM structures of TFIID derived by Eva Nogales (Louder et al., 2016, Patel et al., 2018) is human TFIID on the *Drosophila* super core promoter (which includes “pausing elements” such as DPE and MTE). For this reason, we believe that the insights obtained from stably paused promoters will be broadly relevant.

We have now extensively rewritten the Introduction to better explain our reasoning behind the choice of promoters. In short, they are easy to study and due to known interactions with TFIID, are relevant for mammalian promoters. Finally, we point out current limitations of the assay in our Discussion at the end.

3) To better support the authors claims, they should perform a genomic analysis with their existing data on the stability of pausing at promoters with various combinations of the TATA and Inr elements. They already have the melanogaster data from Kc cells +/- triptolide and should have plenty of sequencing depth given the number of plasmids analyzed. Thus, in theory, this would simply require analysis of the genomic reads without additional experimentation. This would serve to corroborate the results and generalize the model to more promoters.

We liked the idea of performing a more combinatorial analysis of the promoter elements with regard to the paused Pol II half-lives, and this approach turned out to be very fruitful. While we had previously analyzed the genome-wide relationship between core promoter elements and paused Pol II half-lives (see Shao et al., 2017), we had not specifically looked at how various combinations correlate with Pol II pausing. This new analysis now revealed that the Inr sequence with a G at the +2 position most strongly correlates with pausing, independently of which other core promoter elements are found in the promoter.

We have now added the results from this new analysis to Figure 4A and Figure 7A with additional information found in Figure 4—figure supplement 1, Figure 7—figure supplement 1 and Table S6 in Supplementary file 1.

4) The claim that the reporter assay recapitulates the genomic assay could also use strengthening. This point is supported by 8 genes in the current manuscript, which is probably not enough for statistically rigorous conclusions. Figure 2—figure supplement 2 shows the 8 examples, and there is quite some variability in them. The question is how much variability, how to quantify it, and whether or not conclusions can be made given this variability. Given that a lot of conclusions are drawn from comparing the shapes of these promoter distributions, I would think the extent to which they can be reliably made is important. dve, the gene selected for the majority of mutational tests, is an example that shows differences between genomic and reporter contexts.One idea is to bin the promoters into, say 10-20bp bins, and compare the normalized read counts across the bins for the reporter and genomic assay. Then the mean squared error or the correlation coefficient can be computed comparing the reporter and the genomic assay, and the degree of variability can be measured across more genes. This would at least give us some context for how much variability can be expected, and if the differences observed due to the mutations are significantly more than that baseline difference. This analysis or something similar is considered critical.

We had a supplementary table (Table S3 in Supplementary file 1) that showed the correlation coefficient between the bp of the Pol II profiles from the plasmid versus those of the genomic region. They range from 0.77 to 0.96 (mean 0.92), which is very high given that we are not even binning the data into 10-20 bp bins (binning would presumably smooth the data and give higher correlations).

To strengthen this point even further, we have now in the revised manuscript compared these correlation coefficients to those obtained between biological replicate experiments on the plasmids (Table S2 and S3 in Supplementary file 1). Overall, we found the correlations between plasmid and genomic context to be only slightly lower than those between replicates (on average 0.86 versus 0.92), suggesting that the plasmid recapitulates the genomic Pol II pausing profile remarkably well despite possible differences in chromatin or regulatory context. We have added this result to the main text (subsection “Promoter-specific Pol II pausing properties are recapitulated on the reporter”, fourth paragraph), as well.

5) The novelty of finding a strong initiator at highly paused genes is severely over-exaggerated and previous literature is inadequately cited. The connection between Initiator consensus elements, highly focused transcription initiation and high levels of paused RNAPII has appeared multiple times in the literature previously, at least as far back as 2008 (Hendrix et al., 2008), and also in 2010 (Gilchrist et al., 2010).Further, pausing has been recapitulated on DNA sequences in vitro, thus we already know that DNA sequence is a major contributor to pausing. See many in vitro studies from the Gilmour lab (some cited) and also Adelman and Lis (2005).Finally, the authors mention in the Discussion that functional data supporting a role for promoter sequences and pausing are missing. While they are correct that there is a dearth of these important functional studies, the Lis lab did a functional analysis pause elements in the Drosophila Hsp70 promoter, by inserting sequences that changed the position of these elements. See Kwak et al., 2013. Given that there are so few studies like this and the level of detail covering other aspects of the literature, I think this experiment is worth discussing in this context.It doesn't take anything away from the authors' accomplishments to adequately cite previous research in a scholarly way. Appropriate citations must be added.

The statistical association between the Inr and Pol II pausing has been described before, by ourselves in collaboration with Mike Levine’s lab (Hendrix 2008) and in Gilchrist et al., 2010 (shown in Figure 2—figure supplement 2, not described in main text). However, this association could have been indirect since another pausing element, the DPE, requires the Inr to function properly (see review by Kadonaga, Dev Biol 2012). Due to the possible indirect effect, while the association between DPE (or PB) and pausing is more evident, the role of the Inr has been much less clear. In addition, the role of the Inr in Pol II pausing is further complicated by the observation that the Inr functions synergistically with the TATA box, which is associated with reduced Pol II pausing.

Regardless of how one interprets these statistical associations, a causal role in Pol II pausing had not been adequately shown for any core promoter element. We believe our results are novel and significant as we provide strong evidence that core promoter elements impact Pol II pausing individually. To the best of our knowledge, such direct functional examinations have not been done before (the spacing experiments by the Lis lab are interesting but do not address which promoter element mediates the observed effects). Furthermore, in the revised manuscript, we not only provide evidence for a direct role of the Inr in Pol II pausing, but we also show that statistically speaking, the G in the Inr plays a dominant role in Pol II pausing, which is quite surprising.

To provide more clarity for the reader, we have now rewritten the manuscript to better describe what was (or was not) known about the Inr, while citing the requested papers (Introduction, subsection “The Inr strongly contributes to the degree of Pol II pausing”, and subsection “The Inr plays an important role in Pol II pausing”). This should help the reader better understand the significance of our finding.

6) Also related to the ambiguity in the number of promoters involved in each analysis: In the section "The Inr strongly contributes to the degree of Pol II pausing", I think it would be enhanced if the text listed the number of promoters examined for each group. For example, the text says "We first analyzed the naturally occurring Inr sequences from TATA-containing promoters versus those of stably paused promoters". I would like to know when reading this how many promoters are in each set leading to the observations e.g. that TATA-containing promoters have fewer Gs at the +2 position. In particular, when reading the caption for Figure 5A, it says n=132, and this implied that the other panels also used 132 promoters. However I expect that the other panels were based on a larger set (at least panel B). Ideally, more than 132 promoters would have been used for this analysis. Overall, added clarity on the number of promoter for Figure 5B and elsewhere would be helpful.

We had previously described this promoter selection in detail in the Materials and methods under subsection“Sequence analysis of TATA promoters and stably paused promoters”. To make it easier for the reader, we now describe the selection of the promoters briefly in the main text (subsection “The Inr strongly contributes to the degree of Pol II pausing”) and the Figure 5 legend. Furthermore, we perform a more general analysis of the combinatorial relationship between core promoter elements and pausing half-life (Figure 4A and 7A), which places the above-mentioned figure into context.

[Editors' note: further revisions were requested prior to acceptance, as described below.]Essential revisions:1) As the authors acknowledge in the Introduction, pausing takes place at all (or nearly all) genes. The duration and stability of pausing varies, but pausing appears to be an obligate part of the transcription cycle. Thus, the statement in the Abstract that "a G at the +2 position, is critical for Pol II pausing", is not accurate. The authors should change this to clarify that they mean 'stable pausing' or something of this sort. But a G and +2 is certainly not critical for the fundamental process of pausing itself.

We agree: in fact, in two out of four sentences we had said “critical for stable Pol II pausing”. We have now changed the remaining two sentences also to “critical for stable pausing”.

2) The authors should note early in the Results section that 1h Triptolide treatment is a very long treatment, and that secondary effects are likely to occur during such extended intervals of transcription inhibition. Most labs use 2-20 minutes of Triptolide treatment to avoid such potential artifacts. We appreciate that the authors must rely on very long treatments to overcome the considerable noise in their assay, but this should be acknowledged as a potential source of artifact, and as a potential limitation to this study. This idea is briefly mentioned in the Discussion, but it must be noted in the Results section to make readers aware of this caveat.

We appreciate the concern about side effects, which we share when using drug treatments. However, we have not observed concerning side effects of treating our cells with triptolide for 60 min (see Shao and Zeitlinger, 2017) and note that many investigators have used such extended treatments:

- The Bentley lab paper cited below used 10-60 min;

- The Shilatifard lab paper (Chen et al.) used a time-course of up to 120 min and performed most of their further analysis on the 60 min timepoint. Like ours, their control gels on polymerase abundance and phosphorylation show no detectable side effects;

- The John Lis lab paper (Jonkers et al.) used 12.5 min, 25 min and 50 min timepoints.

We did not mean to convey the impression in our last response letter that our assay has “considerable noise”. As can be seen from the small error bars in the figures, biological replicates are very reproducible, even when treating for 5 min (Figure 3). Shorter triptolide treatment times may have more noise than longer treatment times, but this could be addressed by performing more replicates. In other words, using 60 min triptolide treatment was not born out of necessity, but a strategic choice, since it allowed us to choose promoters with maximal differences in Pol II pausing stability (and we had not encountered variability or other concerning side effects with 60 min treatment times).

To be sure, we now mention the better accuracy for longer Pol II half-lives in the Results section: “… we found these measurements to highly reproducible across biological replicates, although the reproducibility is somewhat lower with shorter treatments times of triptolide, presumably because small experimental fluctuations have a larger effect.”

3) The Bentley lab has recently questioned the interpretation of Triptolide data (in particular that using ChIP-based strategies such as in Shao et al. 17), noting that at some genes the Triptolide-mediated block to initiation traps Pol II at the TSS. This effect is evident at the Pino and pepck genes in this study. The authors should cite the recent work from the Bentley lab, and explain whether their ChIP-nexus assay gives them the spatial resolution needed to separate paused from un-initiated Pol II. This will help place the current work in a proper context, and will be clarifying for the field.

We agree that it is curious that this paper seems to suggest that the long Pol II half-lives we observe are due to us measuring Pol II at the initiation site. As we clearly describe in our paper, the high resolution of ChIP-nexus allows us to distinguish paused Pol II from Pol II at the initiation site (Shao and Zeitlinger, 2017). While we observe increased Pol II at the site of initiation after triptolide treatment (this is not a novel finding in the Bentley lab paper), we are specifically measuring the exponential decay of paused Pol II at the site of Pol II pausing to determine the half-life of paused Pol II (the position of pausing is determined based on Pol II pausing after flavopiridol treatment).

To make this clear in the present paper, we have made the following changes:

- Introduction: “We recently performed such time-course measurements of paused Pol II across the *Drosophila melanogaster* genome using a high-resolution exonuclease-based ChIP-seq protocol (ChIP-nexus). This assay has the advantage that it distinguishes between Pol II at the site of initiation and pausing, and thus the Pol II half-life calculations are based on paused Pol II, rather than total Pol II at the promoter (Shao and Zeitlinger, 2017).”

- When we first describe the Pol II profiles after triptolide treatment, we now point out the increased Pol II at the initiation site and cite our paper and the Bentley lab paper: “After 1 hour triptolide treatment, promoters with a Pol II half-life of ~5-10 min show strongly reduced Pol II pausing at the pause position and often show increased levels of Pol II at the site of initiation as previously observed (Shao and Zeitlinger, Erickson et al., 2018). In contrast, stably paused promoters maintain high levels of Pol II at the pausing position with no noticeable increase of Pol II at the site of initiation.”

4) Based on the limited number of experiments aimed at testing the role of downstream promoter sequences, and the limitations in interpreting such experiments, the authors should be more circumspect in the conclusions drawn from these studies. We find it to be an over-reach that the authors conclude that downstream sequences affect pausing less than expected. Given the lack of data to this point, it is safest to note that this wasn't rigorously evaluated, and leave it at that.

We have now removed this section from the Discussion.

5) We understand that the authors went into this study expecting the TATA box to elicit a strong effect on pausing, and that they were surprised that this is not the case. However, the large amount of inconclusive data presented on this topic, and the discussion thereof, is confusing. If the main conclusion is that TATA doesn't seem to have a strong effect on pause stability, and that any effect is highly context dependent and easily over-ridden by other sequences, then there is a much more straightforward way to present this. We strongly suggest that the authors simplify the presentation of the TATA data to focus on the main conclusion, rather than enumerating all the inconclusive or contradictory findings they obtained.

We have now extensively revised the text for clarity and have taken out any unnecessary commentary or discussion of the effect of TATA (we also removed TATA from the Abstract). Instead, we focused on the role of the Inr.

6) Given that computational assays have routinely defined the Inr consensus to have a G at +2, should this really be called the Inr-G variant? Instead, we recommend that this be considered the consensus. It should be clarified that this is the 'norm' and not something that deviates from the norm.

That is a good question, but based on biochemical studies, the *Drosophila* Inr consensus motif is TCAKTY, where K stands for T or G, and both the T and G at +2 are equally functional. Furthermore, the G is not over-presented among mammalian promoters, suggesting that the G is not the better nucleotide for Pol II initiation *per se*. The reason computational studies have predominantly found the G variant is presumably because the Inr-G variant is less degenerate, easier to identify and shows increased conservation. Therefore, we believe it is important to point out and emphasize the presence of the G at the +2 position and feel that it conveys the information more precisely.

7) The discussion of pausing in its genomic context versus on a plasmid at the beginning of the Discussion is difficult to understand in light of the literature.First, there have been a number of studies demonstrating that stable pausing does not involve downstream nucleosomes. The clearest of these demonstrations was Li and Gilmour, EMBO J, 2103). This paper should be cited and the authors should edit the first paragraph to reflect this work, and the fact that it is not at all surprising that pausing could take place on a plasmid or outside of the chromatin context.

Actually, there are many papers supporting a role for nucleosomes in Pol II pausing, but we agree that this is not a resolved issue and the field is quite split. Since this is not an important point in the paper, we therefore decided to take out this entire section from the Discussion.

8) It was demonstrated many years ago that pausing is rarely inhibitory in its endogenous context, because of the positive effect of the paused polymerase on maintaining open chromatin (Gilchrist G and D 2008). In this study, it was specifically demonstrated that the positive effect of pausing was clear in the endogenous context, but did not occur on a plasmid. Thus, it has been previously established that the effect of pausing on gene output is different on a plasmid than in the genome, and that any minor inhibitory effect seen on a plasmid is NOT borne out in the endogenous locus. Thus, it is perplexing that the authors claim that pausing is inhibitory based on their reporter assay- since this is known to be an inaccurate read-out of endogenous activity. Thus, the comments in the Discussion suggesting that the +2 G version of Inr would be inhibitory to transcription because it promotes pausing are unfounded and should be removed.

We have removed this part of the Discussion as well.